# The effects of the sex chromosomes on the inheritance of species-specific traits of the copulatory organ shape in *Drosophila virilis* and *Drosophila lummei*

Alex M. Kulikov[1], Svetlana Yu. Sorokina[1], Anton I. Melnikov[1], Nick G. Gornostaev[1], Dmitriy G. Seleznev[2], Oleg E. Lazebny[1]*

1 Department of Evolutionary Genetics of Development, Koltzov Institute of Developmental Biology of the Russian Academy of Sciences, Moscow, Russia, 2 Department of Ecology of Aquatic Invertebrates, Papanin Institute for Biology of Inland Waters of the Russian Academy of Sciences, Borok village, Yaroslavl Region, Russia

* oelazebny@gmail.com

**Data Availability Statement:** All relevant data are within the manuscript and its Supporting information files.

## Abstract

The shape of the male genitalia in many taxa is the most rapidly evolving morphological structure, often driving reproductive isolation, and is therefore widely used in systematics as a key character to distinguish between sibling species. However, only a few studies have used the genital arch of the male copulatory organ as a model to study the genetic basis of species-specific differences in the *Drosophila* copulatory system. Moreover, almost nothing is known about the effects of the sex chromosomes on the shape of the male mating organ. In our study, we used a set of crosses between *D. virilis* and *D. lummei* and applied the methods of quantitative genetics to assess the variability of the shape of the male copulatory organ and the effects of the sex chromosomes and autosomes on its variance. Our results showed that the male genital shape depends on the species composition of the sex chromosomes and autosomes. Epistatic interactions of the sex chromosomes with autosomes and the species origin of the Y-chromosome in a male in interspecific crosses also influenced the expression of species-specific traits in the shape of the male copulatory system. Overall, the effects of sex chromosomes were comparable to the effects of autosomes despite the great differences in gene numbers between them. It may be reasonably considered that sexual selection for specific genes associated with the shape of the male mating organ prevents the demasculinization of the X chromosome.

## Introduction

Reproductive isolation contributes to the evolutionary process by allowing diverging species to accumulate genetic variation independently, including for adaptively important traits. Isolation is ensured by pre- and postzygotic isolating mechanisms, which differ in evolutionary origin and physiological basis. Postzygotic isolation arises as independent genetic variations

**Funding:** This work was conducted under the Institute of Developmental Biology Russian Academy of Sciences Government basic research program, No. 0108–2019-0007. The funders had no role in study design, data collection and analysis, decision to publish, or preparation of the manuscript.

**Competing interests:** The authors have declared that no competing interests exist.

accumulate in isolated populations and cause sterility and/or non-viability of hybrid progenies because of nonadditive interactions of the alleles at various loci. In contrast, each allele does not exert a deleterious effect on the gene pool of its original population [1–6]. Prezygotic isolation results mainly from the variation driven by sexual selection and prevents the mating of individuals from different populations possessing different adaptations. In allopatric speciation, the selection of traits acting at the copulation stage starts when postzygotic incompatibility has already formed. Reinforcement [7,8] and the sexual selection or sexual conflict [9–12] mechanisms can be involved in this case. In sympatric speciation, a prezygotic barrier should start forming before postzygotic isolating mechanisms are involved. It is noteworthy that experimental findings agree with the assumption that prezygotic isolation accumulates at a higher rate as compared with postzygotic isolation. Coyne and Orr [13,14] estimated the parameters in experiments with many closely related Drosophila species and confirmed a higher accumulation rate for prezygotic isolation.

Selection affects sex chromosome variation differently during the formation of prezygotic versus postzygotic isolation. The fixation of new alleles, which is essential for incompatibility to arise, implies the effect of positive selection only. Postzygotic incompatibilities are not selected directly, but, functional associations of particular genes with viability and fertility traits mediate their choice. The sex chromosomes are enriched in sexually antagonistic genes [15,16] and may more often be targeted by selection at differentially expressed traits. Genes involved in gametogenesis act as targets in the case of postzygotic isolation; and genes involved in mating behavior act as targets in the case of prezygotic isolation. In species with XY sex determination, the X chromosome, as a rule, undergoes demasculinization [16,17], while postzygotic isolation mechanisms act more often in males, in agreement with Haldane's rule. The dominance theory [6,18,19] postulates that recessive variations of X-chromosomal incompatibility loci are expressed in hemizygous males and explains well a viability decrease unrestricted to sex-linked traits. The discrepancy between X-chromosome demasculinization and fertility impairment in males can be explained by nonadditive interactions of alleles and, primarily, recessive negative epistasis. Epistatic interactions usually ensure homogametic sex-limited expression and are altered in a hybrid genome, leading to aberrant expression in the heterogametic sex to cause infertility and a drop-in fitness [16].

Following the drive hypothesis, selection may additionally act indirectly, by suppressing sex chromosome meiotic drive, which results from sex chromosome evolution and permanent internal conflict between sex determination systems [20–23]. For example, various systems of sex chromosome segregation distorters have been identified in the X chromosome and autosomes in *Drosophila* species by an incompatibility locus analysis; the systems include pseudogenes, repeat regions [21,22,24], and heterochromatin blocks responsible for the failure of sister X chromosomes to segregate during mitosis in the hybrid genome [25,26].

The role that the X chromosome plays in the selection-driven formation of prezygotic isolation depends to a substantial extent on the sex wherein a particular trait is expressed given that the X chromosome experiences demasculinization and a direct effect of positive selection, X-chromosome variation is mostly restricted to female choice traits [27]. However, demasculinization of the X chromosome is not absolute, and the hemizygous status of the X chromosome dramatically increases the effect of selection in males [28,29], while not excluding an accumulation of alleles associated with particular characteristics of male mating behavior. Asymmetry of male success in reciprocal interspecific crosses of related species [30–32] formally resembles Haldane's rule. However, it differs in arising directly at the stage of choosing a partner from a related species, being based on the differences accumulated in the respective genomes. Unequal partner preferences in reciprocal crosses depend on the reaction norm of each of the

species concerning partner choice traits [33] and do not suggest linkage with the sex chromosomes.

In *Drosophila*, the X chromosome has been shown to play a role in several species-specific features that are observed in male mating behavior and prevent heterospecific mating, the set including various elements of the courting ritual and various modalities [34–43]. The results have shown that all chromosomes, including the sex chromosomes, contribute to the species-specific variation in mating song elements, while the set of chromosomes or loci associated with the differences vary among related species pairs. The variation due to interspecific differences displayed no predominant association with the sex chromosomes [44].

Copulation, as a final stage of mating behavior, determines the efficiency of the insemination of the female and the contribution of the male genotype to the gene pool of the next generation. The copulation efficiency depends on the copulation duration and the insemination reaction. Male accessory gland secreted proteins (Acps), which evolve at a far greater rate as compared with hemolymph proteins [45–50], play a crucial role in the latter. A total of 46 genes are known to code for Acps in *D. melanogaster*, and only six of them are on the X chromosome. A decrease in female fertility in heterospecific crosses of *D. virilis* and *D. americana* is associated with three *D. americana* and two *D. virilis* recessive autosomal genes [51]. The finding agrees with the idea that isolating barriers arise rapidly from a divergence of a few genes and demonstrates that the X chromosome is weakly involved in the formation of prezygotic isolating barriers.

The copulation duration directly depends on how well the male copulatory system matches the female genitalia in shape. Jagadeeshan and Singh [52] observed that in crosses between the closely related species *D. melanogaster* and *D. simulans*, mating behavior was directly associated with the species-specific shapes of the posterior lobes of the genital arch and the cercus, which are external structures of the *Drosophila* male copulatory system. The time course of copulation stages depended on the mechanic coupling of the male and female genitalia, and the coupling depended on the male external genital structures. The final stage of tight genital coupling was 2–5 times shortened in heterospecific crosses, while the preceding unstable coupling stage was prolonged. The findings indicate that female insemination may be incomplete in heterospecific crosses and that a particular mechanism exists to allow mating flies to be sensitive to the specificity of their contact. Laurie and colleagues analyzed the genetic basis of species divergence to the shapes of male copulatory system elements in *Drosophila*. A series of studies was carried out with *D. simulans* and *D. mauritiana* to genetically analyze the species-specific distinctions in the shape of the epandrium, which is an external male genital structure, and at least eight linkage groups involved were localized to the X chromosome and chromosomes 2 and 3 [53–56]. Additive interactions were mostly observed for the respective loci.

The shape of the male genitalia is the most rapidly evolving among all morphological characters, and its control system is a target of sexual selection [57]. However, the shape of the epandrium, which is the posterior lobe of the genital arch, has been used as the only model to study the genetic basis of species-specific differences in the shape of the copulatory system in *D. simulans* and *D. mauritiana*. Do the X chromosome and autosomes play the same role in species-specific traits related to the shape of the copulatory system in other species groups? Does the Y chromosome contribute to the inheritance of these traits? In this study, we used interspecific crosses between *D. virilis* and *D. lummei* and backcrosses of F1 hybrid males with *D. virilis* females to evaluate the contribution of the sex chromosomes and the total contribution of the autosomes to the degree of dominance of the *D. virilis* and *D. lummei* phenotypes in traits related to the shape of the male copulatory system.

## Methods

Strains *D. lummei* 200 (Serebryanyi Bor, Moscow) and *D. virilis* 160 were from the *Drosophila* collection of the Koltzov Institute of Developmental Biology. The latter strain carried the following recessive autosomal markers: *broken* (*b*) on chromosome 2, *gapped* (*gp*) on chromosome 3, *cardinal* (*cd*) on chromosome 4, *peach* (*pe*) on chromosome 5, and *glossy* (*gl*) on chromosome 6. The first two markers are expressed as gaps in the second transverse and L2 veins, respectively; the other markers determine different eye colors, which are well-identifiable visually. The eye color in *D. virilis* wild-type is middle red wine, and it is darker than the eye color of *D. melanogaster* wild-type. The recessive mutation *cardinal* generates in homozygotes of *D. virilis* vermilion-like eyes. The recessive mutation *peach* generates in homozygotes of *D. virilis* yellowish pink eyes. The combination of homozygotes *cardinal* and *peach* produces reddish-brown eyes that darken to brown with age. The most complicated eye mutation is the recessive mutation *glossy* that produces eyes with the surface with an oily luster surface in reflected light and irregular ommatidia. The eye color is slightly redder and brighter. The combination of a homozygote for *glossy* with the other mutations or their combinations leads to the appearance of an oily sheen effect and a violation of the ommatidia shape. $F_1$ hybrid males were obtained from direct crosses of *D. virilis* females with *D. lummei* males and reciprocal crosses of *D. lummei* females with *D. virilis* males. Also, $F_b$ males were obtained by backcrossing hybrid males with *D. virilis* females. The crossing scheme is shown in Fig 1.

Backcross males heterozygous at all autosomes and fully homozygous backcross males were selected for further analysis. Backcrosses with heterozygous $F_1$ males were performed to exclude autosomal recombination events and to allow rigorous analysis of the effects for variants of homozygous and heterozygous combinations of autosomes of the parental species.

All crosses were carried out at 25°C; a standard food medium and glass tubes of 22 mm in diameter were used; the progeny density was 50–70 flies per tube. All males taken for the copulatory organ dissection were virgins and seven days old.

### Analysis of morphological structures

The mating organ was dissected using thin steel needles in a drop of water under a binocular microscope at a magnification of $12 \times 8$. Preparations were incubated in boiling 10% NaOH to remove adipose structures.

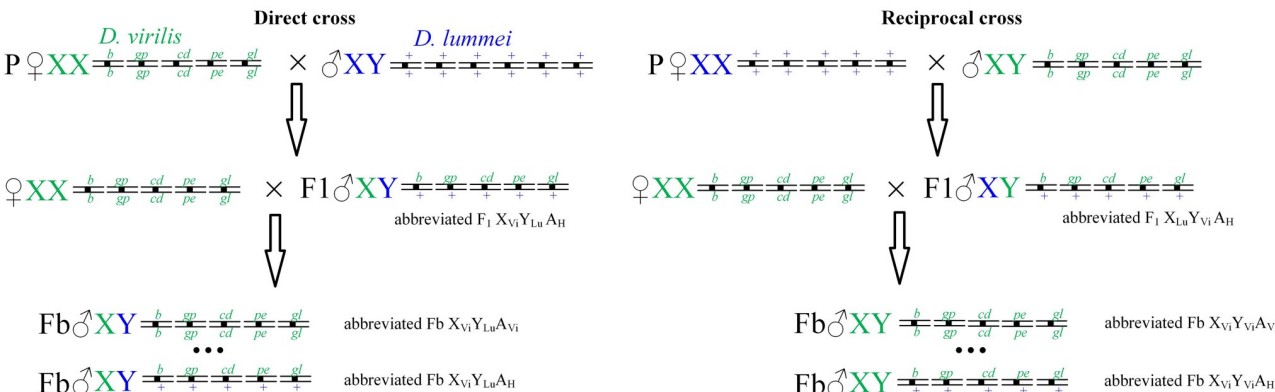

**Fig 1. The scheme of direct (*D. virilis* × *D. lummei*) and reciprocal (*D. lummei* × *D. virilis*) crosses.** Backcrosses were accomplished by the crossing of $F_1$ hybrid males with *D. virilis* females. Chromosomes of *D. virilis* are designated with green, and chromosomes of *D. lummei* are designated with blue. The three dots in Fb males designate all possible intermediate autosome genotypes between the full homozygotes for the marker mutations (*b/b gp/gp cd/cd pe/pe gl/gl*) and full heterozygotes for the marker mutations (*b/+ gp/+ cd/+ pe/+ gl/+*).

A total of 136 preparations were examined, including 14 from *D. lummei* males, 13 from *D. virilis* males, 25 from $F_1$ males obtained by crossing ♀ *D. virilis* × ♂ *D. lummei*, 31 from $F_1$ males obtained by crossing ♀ *D. lummei* × ♂ *D. virilis*, 10 from $F_b$ males with the genotype $X_{Vi}Y_{Vi}$, $A_{Vi} A_{Lu}$, 28 from $F_b$ males with the genotype $X_{Vi}Y_{Lu}$, $A_{Vi} A_{Lu}$, 5 from $F_b$ males with the genotype $X_{Vi} Y_{Vi}$, $A_{Vi} A_{Vi}$, and 10 from $F_b$ males with the genotype $X_{Vi} Y_{Lu}$, $A_{Vi} A_{Vi}$.

As opposed to discretely identifiable phenotypes, such as eye color or vein gaps, the object of quantitative genetics is phenotypes that vary continuously: every character that can be measured (the length or width of the organ or its part) or counted (the bristle number). Frequently, quantitative genetics is referred to as the genetics of complex traits and is based on a model in which many genes influence a trait in a complicated way when they interact within themselves (allele interactions: dominant-recessive relationship of alleles), between themselves (epistatic relationships), and with environmental factors (genotype-environmental interaction) as the model usually accounted for non-genetic factors. According to the current findings, the genetic underpinning of such complex traits is even more complicated as noncoding DNA sequences like enhancers, silencers, and insulators affect the quantitative variability or heritability of quantitative traits. Moreover, such non-genetic effects as epigenetic factors (CpG-islands) may also affect quantitative trait variability. There are different approaches to deal with morphology (traditional morphology when a complex trait is described by a set of measurements or counts, or geometric morphometrics when a set of landmarks or semi-landmarks describes a complex trait) at the stage of collecting data and with the total variability of a complex trait at the stage of statistical analysis. The latter includes a set of different statistical methods beginning from the methods invented by Fisher and Wright based on the decomposition of the total variance, like the analysis of variance, etc., and ending with methods of multivariate statistics and methods of data dimensionality reduction.

Morphometry was carried out using organ images, which were obtained using a Jen-100C electron microscope in the scanning mode at an accelerating voltage of 40 kV and an instrumental magnification of 300–500×. The sagittal view of the phallus was conventionally divided into four areas: the *aedeagus* body, gonites, apodeme, and hook. A coordinate grid was superimposed on the image. A conventional point at the junction of the *aedeagus* area, gonites, and apodema (referred to as the central point) was used as a landmark. Morphometric parameters (MPs) were obtained as distances between the intercrosses of coordinate lines with each other and the image outline. A scheme of measurements is shown in Fig 2; MP characteristics are summarized in Table 1. The declinations of the hook and apodeme (axes MP 2 and MP 28) from axis MP 1 were expressed in radians and designated α and β, respectively. To exclude the dimensionality factor, MP indices (MPIs) were calculated according to a method used previously to evaluate the inter- and intraspecific variations of the genitals in the *virilis* species group (1). MPIs were obtained as MP-to-MP 1 ratios and were numbered according to the numbers of respective MPs. The traits expressed in radians were included in the analyses without normalization.

## Statistical analyses

Statistical analyses were carried out using the program IBM SPSS Statistics v. 23 and the R 3.3 statistical analysis environment with the packages "vegan", "lmPerm", "psych", and "rcompanion". Trait dominance in interspecific crosses was evaluated by comparing the trait variance in fly samples from the parental and hybrid strains; backcross samples included only males with a fully heterozygous autosome set. Differences were tested for significance by multivariate ANOVA; groups displaying homogeneity of variance were identified for each trait by *post-hoc* comparisons, using the Gabriel test and Tukey's HSD test with the Bonferroni correction.

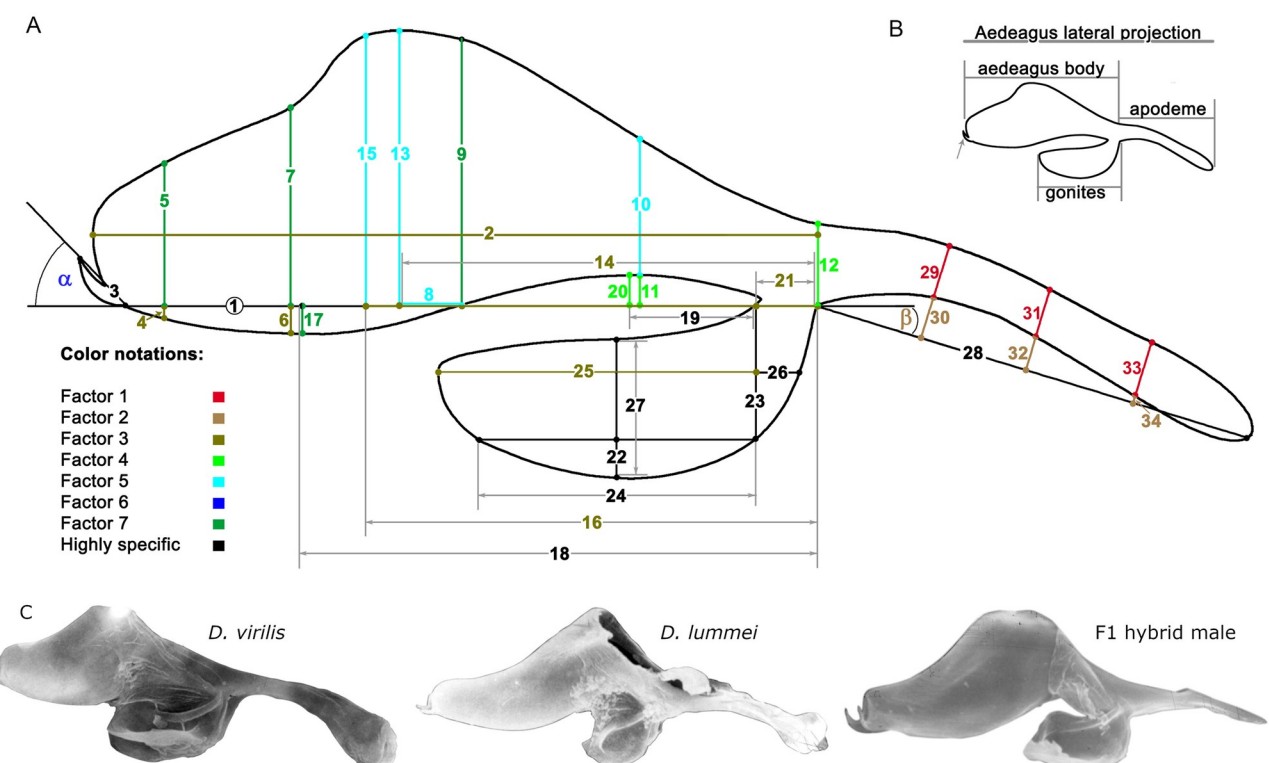

**Fig 2. Outline of the copulatory system with a suite of morphometric parameters (MP) measured in males of the *Drosophila virilis* and *D. lummei* species, F₁, F₂ of the interspecies crosses and backcrosses.** (A) Line segments with corresponding numbers of the same color are grouped according to their weight in a particular latent factor. The color-coding notation of the first seven latent factors and a group of highly specific MPs is located in the lower left-hand corner of the drawing. The number of the main MP1 is placed in a circle. The angle's alpha and beta indicate the angles of deviation of the hook and apodeme from the copulatory organ's central axis. (B) The main parts of the male copulatory organ are shown in the inset in the upper right-hand corner. (C) Photos of the male copulatory organ in *D. virilis*, *D. lummei*, and their hybrid.

The effects of the sex chromosomes, autosomes, paternal genotype, and their interactions on the traits were evaluated using samples from the parental, F₁, and Fᵦ populations, including both homozygotes and heterozygotes for the autosome set. A linear combination of weighted genetic factors was used as a model. The formal model is as follows:

$$X_{ij} = \mu_{+A1E}X \rightarrow Y(\text{epist})_{+A2E}\text{AUT}(\text{add})_{+A3E}P \rightarrow Y + P \rightarrow \text{AUT}(\text{add})_{+A4E} \quad (1)$$

$$X \rightarrow \text{AUT}(\text{rec.epist}) + Y \rightarrow \text{AUT}(\text{rec.epist})_{+A5E}Y \rightarrow_{+A6E}P \rightarrow X +$$

$$P \rightarrow \text{AUT}(\text{dom})_{+A7E}X \rightarrow \text{AUT}(\text{dom.epist}) + X + \text{AUT}(\text{dom})_{+\varepsilon,}$$

where $X_{ij}$ is the trait value in the sample of a particular genotype; $\mu$ is the mean; $\varepsilon$ is the fraction of variance unexplained; $A_n$ values are the weights or indicator variables corresponding to the regression coefficients $E^n$; and n indicates the respective genetic factor or factor combination: X and Y, independent effects of the respective sex chromosomes; AUT(add), the additive effect of recessive autosomal alleles; and AUT(dom), the dominant effect of the autosomes. Pairwise interactions of the factors were also considered, including X→Y(epist), the epistatic effect of the interaction of the X and Y chromosomes; AUT(add) (or AUT(dom))_P, the epigenetic effect of the paternal genotype on the additive or dominant effect of the autosomes; Y (or X)→ AUT(rec.epist), the epistatic effect of the Y (or chromosome on the recessive autosomal alleles; Y (or X)→AUT(dom.epist), the epistatic effect of the Y (or X) chromosome on the autosomal

**Table 1. Description of the morphometric parameters.**

| MP# | Morphometric characteristics |
|---|---|
| 1 | *Aedeagus* length to the hook base |
| 2 | Maximum *aedeagus* length |
| 3 | Hook length |
| 4 | Distance from the MP1 axis to the lower outline as measured 93.75% of the MP1 length away from point 1 |
| 5 | *Aedeagus* height measured 93.75% of the MP1 length away from point 1 |
| 6 | Distance from the MP1 axis to the lower outline as measured 75% of the MP1 length away from point 1 |
| 7 | *Aedeagus* height measured 75% of the MP1 length away from point 1 |
| 8 | Distance from the MP1 axis to the lower outline as measured 50% of the MP1 length away from point 1 |
| 9 | *Aedeagus* height measured 50% of the MP1 length away from point 1 |
| 10 | *Aedeagus* height measured 25% of the MP1 length away from point 1 |
| 11 | Distance from the MP1 axis to the lower outline as measured 25% of the MP1 length away from point 1 |
| 12 | *Aedeagus* height at the base |
| 13 | Maximum distance from the MP1 axis to the upper outline |
| 14 | Distance from the *aedeagus* base to the projection of the highest point of the outline onto the MP1 axis |
| 15 | Distance from the MP1 axis to the elbow of the upper outline |
| 16 | Distance from point 1 to the projection of the elbow point of the upper outline onto the MP1 axis |
| 17 | Distance from the MP1 axis to the lowest point of the *aedeagus* outline |
| 18 | Distance from point 1 to the projection of the lowest point of the *aedeagus* outline onto the MP1 axis |
| 19 | Distance from the projection of the upmost point of the ventral part of the *aedeagus* outline to point 1 |
| 20 | Distance from the MP1 axis to the upmost point of the ventral part of the *aedeagus* outline |
| 21 | Gonite width at the base |
| 22 | Curve depth of the ventral part of the paramere |
| 23 | Distance from the MP1 axis to the paramere outline as measured at the base |
| 24 | Paramere length at the ventral curvature |
| 25 | Paramere length in the central part (at the MP23 midpoint) |
| 26 | Distance from the MP23 midpoint to the outline of the basal part of the paramere (characterizes the paramere bending angle) |
| 27 | Paramere height in the central part (at the MP26 midpoint) |
| 28 | Apodeme length |
| 29 | Apodeme width measured 25% of the MP28 length away from point 1 |
| 30 | Apodeme bending parameter (25% of the MP28 length away from point 1) |
| 31 | Apodeme width measured 50% of the MP28 length away from point 1 |
| 32 | Apodeme bending parameter (50% of the MP28 length away from point 1) |
| 33 | Apodeme width measured 75% of the MP28 length away from point 1 |
| 34 | Apodeme bending parameter (75% of the MP28 length away from point 1) |

dominant alleles; and P→X, the possible nongenetic effect of the paternal genotype on the effect of the X chromosome. A plus sign indicates a combination of the respective effects. The effect of the paternal genotype alone on the traits under study was not considered because nongenetic effects were expected only for a combination of this factor with the offspring genotype. The genetic factors were treated as categorical variables; their significance was assessed by permutational ANOVA (PermANOVA). A paired permutation test was employed in *post-hoc* comparisons, and the Bonferroni correction was used for multiple comparisons. The maximal number of iterations was 100 000; the significance level was 0.05.

Several effects were combined in one variable because the study was not designed to examine all possible combinations of the factors and because several vectors determining the $A_n$

**Table 2. Indicator variables used to obtain the coefficients of regression between hereditary factors or their combinations and trait values.**

| Strain | | D.lum. | D.vir. | $F_1 X_{Lu}Y_{Vi}$ | $F_1 X_{Vi}Y_{Lu}$ | $F_bX_{Vi}Y_{Vi}, A_H$ | $F_bX_{Vi}Y_{Vi}, A_{Vi}$ | $F_bX_{Vi}Y_{Lu}, A_H$ | $F_bX_{Vi}Y_{Lu}, A_{Vi}$ |
|---|---|---|---|---|---|---|---|---|---|
| **Chromosome and paternal statuses Regressors** | X | B | A | B | A | A | A | A | A |
| | Y | B | A | A | B | A | A | B | B |
| | AUT | B | A | H | H | H | A | H | A |
| | P | Lu | Vi | Vi | Lu | H | H | H | H |
| $X{\rightarrow}Y(epist)$ | $A_1$ | 0 | 1 | 0 | 0 | 1 | 1 | 0 | 0 |
| $AUT(add)$ | $A_2$ | -1 | 1 | 0 | 0 | 0 | 1 | 0 | 1 |
| $P{\rightarrow}AUT(add)+P{\rightarrow}Y$ | $A_3$ | -1 | 2 | 1 | 0 | 0 | 0 | 0 | 0 |
| $Y{\rightarrow}AUT(rec.epist)+ X{\rightarrow}AUT(rec.epist)$ | $A_4$ | 0 | 2 | 0 | 0 | 0 | 2 | 0 | 1 |
| $Y{\rightarrow}AUT(dom.epist)+Y$ | $A_5$ | 0 | 2 | 2 | 0 | 2 | 2 | 0 | 0 |
| $P{\rightarrow}X+P{\rightarrow}AUT(dom)$ | $A_6$ | 0 | 2 | 1 | 2 | 0 | 0 | 0 | 0 |
| $X{\rightarrow}AUT(dom.epist) +X+AUT(dom)$ | $A_7$ | 0 | 3 | 1 | 3 | 3 | 3 | 3 | 3 |

*$A_{1-7}$ are the indicator variables according to the model (1). B, the genotype is hemizygous for the sex chromosomes or homozygous for the D. lummei (D.lum. or Lu) autosomes; A, the genotype is hemizygous for the sex chromosomes or homozygous for the D. virilis (D.vir. or Vi) autosomes; H, the genotype is heterozygous for the D. lummei/D. virilis autosomes (AUT) and the paternal (P) genotype is that of an interspecific hybrid; Lu and Vi, the paternal genotype is that of a parental species (D. lummei or D. virilis). Regressors: $X{\rightarrow}Y(epist)$, an epistatic effect of the Y chromosome on the X chromosome; $AUT(add)$, an independent additive effect of recessive autosomal alleles; $P{\rightarrow}AUT(add)+P{\rightarrow}Y$, an epigenetic effect of the homozygous paternal genotype on the additive effect of the autosomes and the effect of the Y chromosome; $Y{\rightarrow}AUT(rec.epist)+X{\rightarrow}AUT(rec.epist)$, an epistatic effect of both of the sex chromosomes on recessive autosomal alleles; $Y+Y{\rightarrow}AUT(dom.epist)$, an independent effect of the Y chromosome and its epistatic effect on autosomal dominant alleles; $P{\rightarrow}X+P{\rightarrow}AUT(dom)$, nongenetic effects of the homozygous paternal genotype on the X chromosome and autosomal dominant alleles; and $X{\rightarrow}AUT(dom.epist) +X+AUT(dom)$, effects of the X chromosome, autosomal dominant alleles, and their epistatic interactions.*

indicator variables were collinear. The values of the corresponding vectors were summed to combine several effects in one pooled effect. Backcrosses to restore the *D. virilis* genotype serve to evaluate the effect of the *D. virilis* chromosomes on the resulting phenotype. The indicator variables were therefore taken to be 1 for the *D. virilis* genotype and 0 for the *D. lummei* genotype when considering effects of the X and Y chromosomes and a dominant effect of the autosomes and to be 1 for homozygotes for the *D. virilis* chromosomes, 0 for heterozygotes, and −1 for homozygotes for the *D. lummei* chromosomes when considering additive effects of the autosomes. The indicator variables for the factor Paternal Genotype were taken to be 1 for homozygous males of the parental species and 0 for heterozygous F1 males to allow for possible spermatogenesis defects and subsequent epigenetic marking of the chromosomes in interspecific hybrids. The resulting matrix of genotype-dependent vector values is shown in Table 2.

A factor analysis was used to capture a covariance of traits; factors were extracted by the maximum likelihood method; factor loadings were examined by rotating principal component axes via the Promax procedure with the Kaiser normalization; estimates were obtained by regression analysis. The number of factors to be extracted isolated was determined using a scree plot of the eigenvalues of the trait correlation matrix.

Because a distinct elbow was not observed in the plot, we chose the factors whose eigenvalues were above a simulation curve obtained by a parallel analysis with 1000 permutations. The results of the factor analysis and their associations with genotypes were visualized by redundancy analysis (RDA), using the aggregate matrix of mean estimates for each factor. A principal component analysis (PCA) was performed for the initial MPs. The first two components were used in visualization.

All stages of the study are reflected in the pipeline of all analyses (Fig 3).

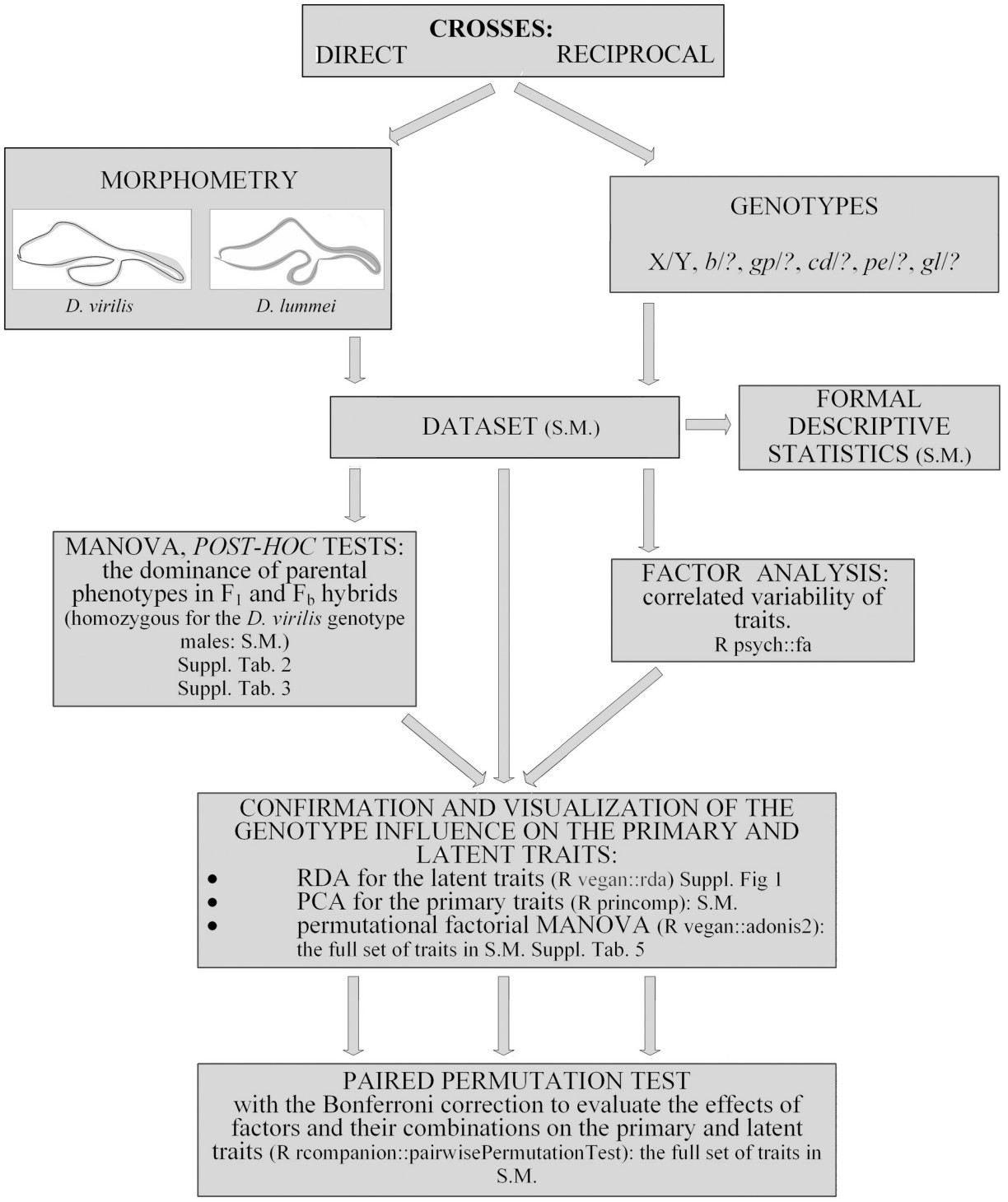

**Fig 3. The pipeline of all analyses in the study.** S.M.: the raw data or the results of the particular analysis are presented in Supplementary Material.

## Results

Following the pipeline (Fig 3) results are presented in the corresponding order with a short introduction to each analysis describing the aim of the particular type of statistical analysis and the significance of the results obtained.

### Correlated variability of traits of the copulatory organ shape

Traits that provide one function are linked by that function, and they should be correlated among them. The aim of the factor analysis is to reveal such correlation pleiads or, in other words, sets of correlated traits and the number of sets corresponding to the number of new latent traits that are called Factors.

To study the structure of correlations among the traits, and the latent factors responsible for a correlated variation, total variance over all samples was examined by factor analysis. Seven factors were taken based on the scree plot and together accounted for 64.4% of the total variance (Table 3). Characteristics of the variation in the traits under study are shown in supplementary materials (S1 Table).

Based on the traits with the greatest factor loadings, the factors were classified as follows.

Factor 1 characterized the variation in the width of the apodeme outline;

Factor 2 characterized the variation in the traits related to the apodeme bend;

Factor 3 included the most distinct species-specific traits and characterized the variation in (a) the shape of the ventral bend in the distal part of the *aedeagus* outline, (b) the position of the dorsal bend of the *aedeagus* outline relative to the central point, and (c) paramere lengths;

Factor 4 characterized the variation in traits related to the ventral bend in the proximal part of the *aedeagus*;

Factor 5 characterized the variation in traits related to the height of the dorsal bend of the *aedeagus* outline;

Factor 6 characterized the variation in the hook angle and the apical part of the *aedeagus* outline; and Factor 7 characterized the variation in traits related to the dorsal bend in the distal part of the *aedeagus* outline and the lowermost point of the ventral bend in the same *aedeagus* part. IMP 3, 18, 19, 23, 24, 26, 27, and 28 were highly specific (Table 3, highly specific).

### Generalized estimate of the dependence of the revealed variability of the phenotype of $F_1$ and $F_b$ hybrids on the genotype

In our case, redundancy analysis (RDA) was carried out to relate the set of variables that determine the factor structure values (a set of explanatory variables) to the variables that

**Table 3. Compositions and eigenvalues of factor loadings as revealed by the maximum likelihood method.**

| Factor 1 (2.674, 7.6) | Factor 2 (3.126, 8.9) | Factor 3 (5.326, 15.2) | Factor 4 (2.021, 5.8) |
|---|---|---|---|
| 29, 31*, 33 | 30*, 32*, 34*, β | 2, 4*, 6*, 14, 16*, 21, 25 | 11*, 12, 20* |
| **Factor 5** (3.084, 8.8) | **Factor 6** (2.476, 7.1) | **Factor 7** (3.412, 9.7) | **Highly specific** |
| 8, 10, 13, 15* | 2, α* | 5, 7*,9*, 17* | 3, 18, 19, 23, 24, 26, 27, 28 |

The eigenvalue and percent variance accounted for by a respective factor are shown in parentheses. Traits included in factor loadings with weights higher than 0.5 are shown.

* indicates the traits that had weights higher than 0.7.

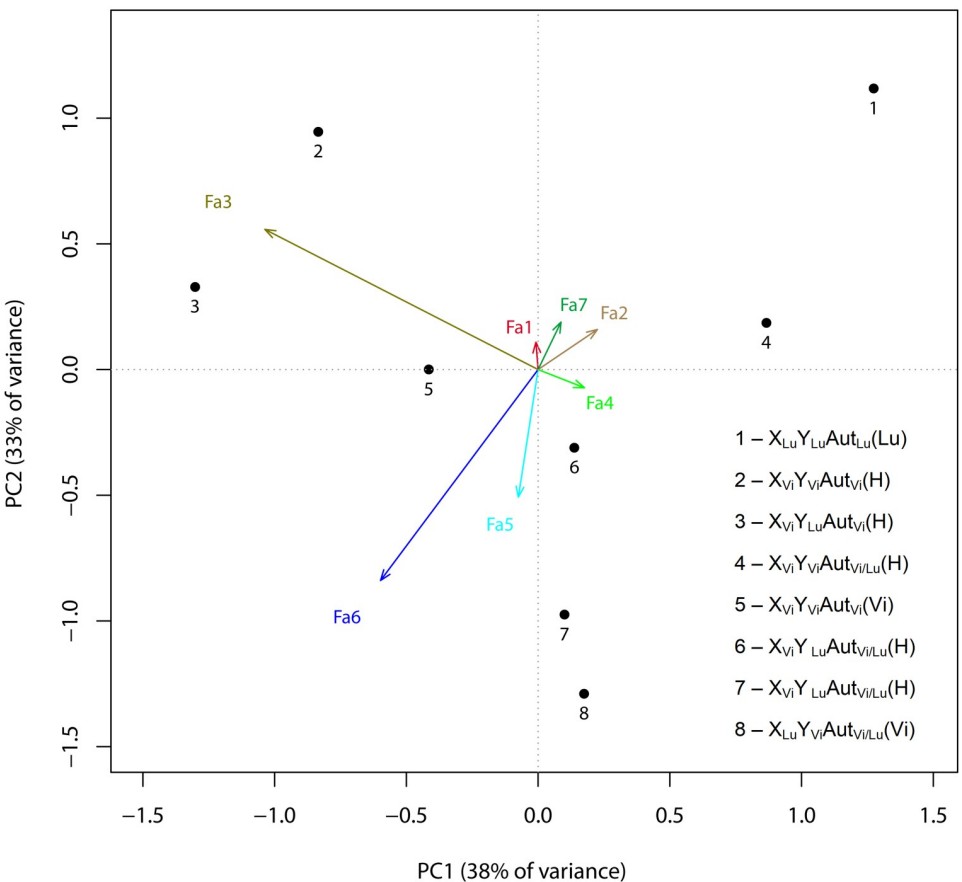

**Fig 4. Redundancy analysis: Association of factor structure values and genotypes.** Numbers denote particular genotypes: 1, $X_{Lu}Y_{Lu}Aut_{Lu}Lu$ (parent male was a *D. lummei* male); 2, $X_{Vi}Y_{Vi}Aut_{Vi}Vi/Lu$ (parent male was a hybrid F1 male with a set of $X_{Vi}Y_{Lu}$ sex chromosomes or with the opposite set of sex chromosomes, $X_{Lu}Y_{Vi}$); 3, $X_{Vi}Y_{Lu}Aut_{Vi}Vi/Lu$ (parent male was a hybrid F1 male with a set of $X_{Vi}Y_{Lu}$ sex chromosomes or with the opposite set of sex chromosomes, $X_{Lu}Y_{Vi}$); 4, $X_{Vi}Y_{Vi}Aut_{Vi/Lu}Vi/Lu$ (parent male was a hybrid F1 male with a set of $X_{Vi}Y_{Lu}$ sex chromosomes or with the opposite set of sex chromosomes, $X_{Lu}Y_{Vi}$); 5, $X_{Vi}Y_{Vi}Aut_{Vi}Vi$ (parent male was a *D. virilis* male); 6, $X_{Vi}Y_{Lu}Aut_{Vi/Lu}Vi/Lu$ (parent male was a hybrid F1 male with a set of $X_{Vi}Y_{Lu}$ sex chromosomes or with the opposite set of sex chromosomes, $X_{Lu}Y_{Vi}$); 7, $X_{Vi}Y_{Lu}Aut_{Vi/Lu}Lu$ (parent male was a *D. lummei* male); 8, $X_{Lu}Y_{Vi}Aut_{Vi/Lu}Vi$ (parent male was a *D. virilis* male).

characterize the genotype (a set of response variables). RDA is a direct gradient analysis technique which summarizes linear relationships between components of response variables that are "redundant" with (i.e. "explained" by) a set of explanatory variables. To do this, RDA extends multiple linear regression (MLR) by allowing regression of multiple response variables on multiple explanatory variables (Fig 4). A matrix of the fitted values of all response variables generated through MLR is then subject to principal components analysis (PCA). In the analysis, the ordination of the genotypes and factors was performed in the space defined by correlations between factor weights, which were weighted sums of trait weights, and linear combinations of genotype scores, which were obtained for the genotypes characterizing the respective point in trait distribution. The center of gravity of the genotype distribution was brought into coincidence with the centers of gravity of the distribution of the factor structures. As seen from Fig 4, a two-dimensional space is defined by the orthogonal factor pair Fa3–Fa4 (X) and Fa6, Fa5–Fa2, Fa7 (Y). The genotypes showed the following arrangement in the two-dimensional factor space: +X: $X_{Vi}Y_{Vi}Aut_{Vi}Vi/Lu$, $X_{Vi}Y_{Lu}Aut_{Vi}Vi/Lu$, +Y($-$X):

$X_{Lu}Y_{Lu}Aut_{Lu}Lu$, $X_{Vi}Y_{Vi}Aut_{Vi/Lu}Vi/Lu$, −Y(− X): $X_{Vi}Y_{Lu}Aut_{Vi/Lu}Lu$, $X_{Lu}Y_{Vi}Aut_{Vi/Lu}Vi$. Genotypes $X_{Vi}Y_{Lu}Aut_{Vi/Lu}Vi/Lu$ and $X_{Vi}Y_{Vi}Aut_{Vi}Vi$ occur in the region of medium values. The angle between vectors on the plot and between genotype positions is proportional to the correlation between them. Therefore, the variation in traits of backcross males homozygous for the autosomes (with the *D. virilis* X chromosome) positively correlates with Factor 3 and negatively correlates with Factor 4. The finding indicates that the Y chromosome insignificantly affects the traits that determine the structures of the two factors. The contribution of the Y chromosome to the individual traits is somewhat greater in the backcross genotypes heterozygous for the autosomes but is still incomparable with the contributions that the Y chromosome makes to the traits in the other genotypes. Positive correlations with Factors 6 and 5 and negative correlations with Factors 2 and 7 were observed for traits of F1 males, opposite correlations, for traits of *D. lummei* males and backcross males with the *D. virilis* sex chromosomes. It is clear that combinations of the sex chromosomes in genotypes heterozygous or homozygous for the *D. lummei* autosomes and the species from which the Y chromosome was inherited (the male parent identity) play a crucial role in the traits involved in the respective factor structures. Fa1–7 are the variation vectors of Factors 1–7, which were obtained by the maximum likelihood method. The variables (traits) that constitute the X-axis (in the order of decreasing factor loading) are: IMP6 (0.922), IMP4 (0.769), IMP16 (0.732), IMP11 (0.918), IMP20 (0.729). The variables (traits) that constitute the second Y-axis are: α (0.890), IMP15 (0.832), IMP32 (1.039), IMP34 (0.9), IMP30 (0.843), IMP17 (0.831), IMP9 (0.739), IMP7 (0.718). The least significant factors are in italics; factor loadings are shown in parentheses. The genotypes are abbreviated: the first letter stands for the *X* chromosome, the second letter stands for the *Y* chromosome, the third letter stands for autosomes, and the fourth letter (in brackets) stands for the parentage of the parent male: *D. virilis* (Vi), *D. lummei* (Lu), or an interspecies hybrid (H); A designates the genotype of *D. virilis*, B designates the genotype of *D. lummei*, H designates the *D. virilis/D. lummei* heterozygote for autosomes and the hybrid status of the parent male. Hybrid parental males already have genomic incompatibility and their sperm can be significantly impaired compared to males from pure species. The chromosomes and paternal genotype are indicated in the following order: the X chromosome, the Y chromosome, autosomes, male parent identity.

The results are supported by the distribution of the genotypes on a scatter plot (S1 Fig) of the two first principal components, which are similar in loading structure to Factors 6 and 3. The parental genotypes are in opposite corners of the plot, having the lowest values in the case of *D. virilis* and the highest in the case of *D. lummei*. The cloud of *D. virilis* data overlaps the clouds of $F_b$ males homozygous for autosomes regardless of the Y-chromosome status. The other genotypes are distributed between the *D. lummei* cloud and the cloud of *D. virilis* with homozygous $F_b$ males; $F_1$ males and $F_b$ males heterozygous for autosomes differ in variation and form slightly overlapping clouds. Different genotypes differ in how close their variation parameters are to the parameters of the parental species at different principal component axes. Again, trait expression depends on the combination of the sex chromosome status, the autosome status, and the origination of the Y chromosome (a male parent identity).

## The dominance of parental phenotypes by quantitative traits of the copulative organ shape of *D. virilis* and *D. lummei* in male offspring of $F_1$ and $F_b$

Two approaches were used to more precisely evaluate the effect that the sex chromosomes exert on trait expression in the shape of the male copulatory system. First, ANOVA was performed to compare the dominance of parental species-specific phenotypes in progenies from

reciprocal crosses between *D. virilis* and *D. lummei* and backcrosses of $F_1$ males with *D. virilis* females. All genotypes had the same autosome set and differed in sex chromosome combination and the identity of the male parent (the original parental species or the $F_1$ hybrid). Second, ANOVA and *post-hoc* tests were performed using the genotype at the sex chromosomes, the genotype at the autosomes, and combinations of these factors, including the male parent identity, as independent variables.

The phenotypes of males obtained in direct and reciprocal crosses and backcross males heterozygous for all autosomes were compared with the phenotypes of males of the parental species. The results are summarized in Table 4. The logic of estimating the degree of dominance for a trait has been described previously [58]. Based on the distribution of the hybrid and parental genotypes over groups isolated by post-hoc comparisons, it is possible to identify the following variants: incomplete dominance, the dominance of the *D. virilis*, or *D. lummei* phenotype, and lack of difference among all phenotypes. The resulting data clustering variants were ranged by the degree of phenotype dominance. All cases where a genotype in question appeared together with the parental genotypes in one group were considered to suggest no significant difference (ns) in evaluating the degree of dominance. The variants f1(b) < l < v, v < l < f1(b), l,f1(b) < v, v < f1(b),l, v ≤ f1(b),l, and l,f1(b) ≤ v suggested the dominance of the *D. lummei* phenotype (DLu); the variants l < f1(b) < v, l ≤ f1(b) ≤ v, v ≤ f1(b) ≤ l, and v < f1(b) < l, intermediate dominance (ID); and f1(b) < v < l, l < v < f1(b), f1(b),v < l, l < f1(b),v, l ≤ v,f1(b), and f1(b),v ≤ l, suggested the dominance of the *D. virilis* phenotype (DVi). The Gabriel test and Tukey's HSD test yielded similar results; only those of the Gabriel test are shown.

IMPs 12, 22, 26, 29, and 31 did not significantly depend on the genotype and are not included in Table 2. The IMPs that showed significant correlations were grouped. We describe the most general results of the analysis of variance. First, traits at which the *D. virilis* phenotype dominated (43) prevailed over traits with the dominance of the *D. lummei* phenotype (32) and traits with intermediate dominance (26) in all four samples. The dominance of the same phenotype regardless of the sex chromosome combination was observed for only 5 (IMPs 7, 16, 27, 28, and 30) out of the 30 IMPs included in the analysis, suggesting a substantial role of the autosome combination and the identity of the male parent in trait expression. The effect of the male parent identity was evaluated by comparing $F_1$ males with genotype $X_{Vi}Y_{Lu}$ and backcross males with the same genotype, given that crossing over is absent in males, and the males in question were entirely identical in chromosome composition. Differences in parental phenotype dominance were observed for 16 out of the 30 traits, and the dominance character changed to the opposite one in the case of apodeme declination angle. Substitution of the *D. lummei* Y chromosome for its *D. virilis* counterpart in backcross males similarly changed the dominance character in 16 traits. Of these, ten traits, which mostly loaded on Factors 1, 3, 4, and 5, showed a change to dominance of the opposite species-specific phenotype relative to the species origin of the Y chromosome. Phenotypic comparisons of $F_1$ progenies showed that simultaneous substitution of the sex chromosomes changed the character of dominance at 12 traits, of which three (IMP 11, IMP20, and β) changed their dominance status to the opposite one, according to the species origin of the X chromosome. Other changes were less distinct and included the following. Firstly, intermediate dominance was shifted toward dominance of one of the parental species. Secondly, the phenotype was changed relative to that of the parental species so that a homogeneous variance group within one of the parental species and offspring with a particular genotype was replaced with a group that included offspring with an alternative genotype and both of the parental species (Table 4, category "ns"). It is of interest to note that the most remarkable difference in the total number of traits showing dominance of one of the parental phenotypes was observed in backcross flies heterozygous for the

**Table 4. The dominance of the copulative system shape-related traits as dependent on the sex chromosome composition in *D. virilis*/*D. lummei* hybrid males, heterozygous for the autosomes.**

| Factor | Sign | ♀$_{Lu}$ × ♂$_{Vi}$ F$_I$ X$_{Lu}$Y$_{Vi}$ | | ♀$_{Vi}$ × ♂$_{Lu}$ F$_I$ X$_{Vi}$Y$_{Lu}$ | | ♀$_{Vi}$ × ♂F$_1$(♀$_{Vi}$ × ♂$_{Lu}$) F$_b$ X$_{Vi}$Y$_{Lu}$ | | ♀$_{Vi}$ × ♂F$_I$(♀$_{Lu}$ × ♂$_{Vi}$) F$_b$ X$_{Vi}$Y$_{Vi}$ | |
| --- | --- | --- | --- | --- | --- | --- | --- | --- | --- |
| | | D$_x$ | P.-h. | D$_x$ | P.-h. | D$_x$ | P.-h. | D$_x$ | P.-h. |
| F1 | IMP33 | D$_{Vi}$ | l<v,f$_1$ | D$_{Vi}$ | l≤f$_1$,v | D$_{Vi}$ | l≤f$_b$,v | ns | f$_b$,l,v |
| F2 | IMP30 | D$_{Lu}$ | f$_1$,l<v | D$_{Lu}$ | l,f$_1$≤v | D$_{Lu}$ | f$_b$,l<v | D$_{Lu}$ | l,f$_b$≤v |
| F2 | IMP32 | D$_{Vi}$ | f$_1$,v≤l | ns | f$_1$,v,l | D$_{Vi}$ | f$_b$≤ v,l | D$_{Vi}$ | f$_b$,v≤l |
| F2 | IMP34 | D$_{Vi}$ | f$_1$<v<l | ns | v,f$_1$,l | D$_{Vi}$ | f$_b$,v<l | ns | v,fb,l |
| F2 | beta | D$_{Lu}$ | l,f$_1$<v | D$_{Vi}$ | l<v,f$_1$ | D$_{Lu}$ | l,f$_b$<v | ID | l≤f$_b$<v |
| F3 | IMP4 | ns | f$_1$,l,v | D$_{Lu}$ | f$_1$,l<v | ns | f$_b$,l,v | D$_{Lu}$ | f$_b$,l<v |
| F3 | IMP6 | D$_{Lu}$ | l,f$_1$<v | D$_{Lu}$ | l,f$_1$<v | ID | l,f$_b$<v | D$_{Lu}$ | l,f$_b$<v |
| F3 | IMP14 | ID | l<f$_1$<v | ID | l<f$_1$<v | ID | l<f$_b$<v | D$_{Lu}$ | l,f$_b$<v |
| F3 | IMP16 | ID | l<f$_1$<v | ID | l<f$_1$<v | ID | l<f$_b$<v | ID | l≤f$_b$<v |
| F3 | IMP21 | ns | l,f$_1$,v | D$_{Vi}$ | l≤f$_1$,v | ns | l,f$_b$,v | ns | l,f$_b$,v |
| F3 | IMP25 | ID | l<f$_1$<v | D$_{Vi}$ | l<f$_1$,v | ID | l<f$_b$<v | D$_{Lu}$ | l,f$_b$<v |
| F4 | IMP11 | D$_{Lu}$ | f$_1$,v<l | D$_{Vi}$ | f$_1$,v≤l | ns | v,f$_b$,l | D$_{Lu}$ | v≤l,f$_b$ |
| F4 | IMP20 | D$_{Lu}$ | v<f$_1$,l | D$_{Vi}$ | v,f$_1$<l | D$_{Vi}$ | f$_b$,v<l | ID | v≤f$_b$≤l |
| F5 | IMP8 | D$_{Vi}$ | l<v,f$_1$ | D$_{Vi}$ | l<v,f$_1$ | D$_{Vi}$ | l<v,f$_b$ | ID | l≤f$_b$≤v |
| F5 | IMP10 | D$_{Lu}$ | l,f$_1$<v | D$_{Lu}$ | l,f$_1$<v | ID | l≤f$_b$≤v | ID | l≤f$_b$≤v |
| F5 | IMP13 | D$_{Vi}$ | l<v,f$_1$ | D$_{Vi}$ | l<v,f$_1$ | ID | l≤f$_b$≤v | D$_{Lu}$ | l,f$_b$<v |
| F5 | IMP15 | D$_{Vi}$ | l≤v,f$_1$ | D$_{Vi}$ | l≤v,f$_1$ | D$_{Vi}$ | l≤v,f$_1$ | ns | f$_b$,l,v |
| F3,6 | IMP2 | D$_{Vi}$ | f1,v<l | D$_{Vi}$ | v,f1≤l | ns | fb,v,l | ns | v,fb,l |
| F6 | alpha | D$_{Vi}$ | l<v<f$_1$ | D$_{Vi}$ | l<v<f$_1$ | ID | l<f$_b$<v | ID | l<f$_b$<v |
| F7 | IMP5 | D$_{Lu}$ | f$_1$,l≤v | D$_{Lu}$ | f$_1$,l≤v | D$_{Lu}$ | f$_b$,l<v | ns | l,f$_b$,v |
| F7 | IMP7 | D$_{Vi}$ | v,f1≤l | D$_{Vi}$ | v,f$_1$≤l | D$_{Vi}$ | v,f$_b$≤l | D$_{Vi}$ | f$_b$,v≤l |
| F7 | IMP9 | D$_{Vi}$ | v,f$_1$≤l | ID | v≤f$_1$≤l | D$_{Vi}$ | v,f$_b$<l | D$_{Vi}$ | v,f$_b$<l |
| F7 | IMP17 | D$_{Vi}$ | f$_1$,v<l | ID | v<f$_1$≤l | D$_{Vi}$ | f$_b$,v<l | D$_{Vi}$ | v,f$_b$<l |
| CS | IMP3 | ID | v≤f$_1$≤l | D$_{Lu}$ | v< f$_1$,l | D$_{Lu}$ | v<l,f$_b$ | D$_{Lu}$ | v<l,fb |
| CS | IMP18 | ID | l≤f$_1$≤v | D$_{Lu}$ | f$_1$,l <v | ID | l≤f$_b$≤v | D$_{Vi}$ | l<v,f$_b$ |
| CS | IMP19 | ID | l≤f$_1$≤v | ID | l≤f$_1$≤v | ID | l≤f$_b$≤v | D$_{Vi}$ | l<f$_b$,v |
| CS | IMP23 | ns | l,f$_1$,v | ns | l,v,f$_1$ | ns | l,v,f$_b$ | D$_{Vi}$ | l≤v,f$_b$ |
| CS | IMP24 | ID | l≤f$_1$≤v | ID | l≤f$_1$≤v | D$_{Lu}$ | l,f$_b$<v | D$_{Lu}$ | l,f$_b$<v |
| CS | IMP27 | D$_{Vi}$ | l≤f$_1$,v | D$_{Vi}$ | l≤f$_1$,v | D$_{Vi}$ | l≤f$_b$,v | D$_{Vi}$ | l≤f$_b$,v |
| CS | IMP28 | D$_{Lu}$ | f$_1$,l≤v | D$_{Lu}$ | l,f$_1$≤v | D$_{Lu}$ | l,f$_1$≤v | D$_{Lu}$ | l,f$_b$≤v |

Traits are grouped according to their maximal weights in the factors extracted isolated by the maximum likelihood method (Table 3); CS, highly specific trait; D$_x$, dominance: D$_{Lu}$, the dominance of the *D. lummei* phenotype; D$_{Vi}$, the dominance of the *D. virilis* phenotype; ID, intermediate dominance; ns, a nonsignificant difference. Crosses and male genotypes are specified in the two uppermost rows. P.-h., results of post-hoc comparisons: f$_1$, f$_b$, l, v are the mean values of a respective trait in the F$_1$ progeny, backcross progeny, and *D. lummei* and *D. virilis* parental males, respectively; the symbols are arranged in the order of increasing trait values. Symbols separated with a comma correspond to a group obtained by pooling samples homogeneous in variance.

autosomes. Thus, F$_b$ X$_{Vi}$Y$_{Vi}$ males had eight traits at which the *D. virilis* phenotype dominated and ten traits at which *D. lummei* phenotype dominated, while F$_b$ X$_{Vi}$Y$_{Lu}$ males displayed an opposite pattern and had ten and five such traits, respectively.

As expected, the number of traits at which the *D. virilis* phenotype dominated increased in backcross males homozygous for the *D. virilis* autosomes (S2 Table); the set included 19 traits in F$_b$ X$_{Vi}$Y$_{Vi}$ males and 23 traits in F$_b$ X$_{Vi}$Y$_{Lu}$ males. Substitution of the Y chromosome changed the character of dominance at nine traits, of which seven again demonstrated an adverse effect of the species identity of the Y chromosome on the phenotype.

**Table 5. Significant effects of chromosomes on the total variability of the traits of the shape of the copulative apparatus and the size of these effects.**

|  | Wilks' α | F | Effect df | Error df | p | Partial eta-squared | Non-centrality parameter, λ | Observed power (α = 0.05) |
|---|---|---|---|---|---|---|---|---|
| **Chr X** | 0.420 | 3.4 | 35 | 86 | 0.000 | 0.580 | 118.9 | 1.000 |
| **Chr Y** | 0.590 | 1.7 | 35 | 86 | 0.024 | 0.410 | 59.7 | 0.986 |
| **AUT** | 0.031 | 11.4 | 70 | 172 | 0.000 | 0.823 | 798.2 | 1.000 |
| **Pmale** | 0.148 | 3.9 | 70 | 172 | 0.000 | 0.615 | 274.4 | 1.000 |

Chr X–chromosome X, Chr Y–chromosome Y, AUT–autosomes, Pmale–nongenetic effect of the parent species (*D. virilis* or *D. lummei*), which Y chromosome was inherited by a male; Wilks' α—a test statistic that is reported in results from factorial MANOVA, F–Fisher's F-statistic, df–degrees of freedom, p–probability value, Partial eta squared is the ratio of variance associated with an effect, plus that effect and its associated error variance, λ–one of the two parameters of the noncentral chi-square distribution, observed power is an estimate of the power of a study based on the observed effect size in a study.

Factorial MANOVA was carried out to evaluate the effects of the sex chromosomes, autosomes, paternal genotype, and their interactions on the phenotypic traits. Significant effects were observed for each of the four factors taken alone and the interaction of the Y chromosome and autosomes (S3 Table).

To study the effect on particular phenotypic traits for each of the factors, factorial MANOVA was performed with a forced incorporation of all four predictors and the effect of the ChrY\*Aut interaction. The results are summarized in S4 Table. The majority of the traits showed an association with the identities of the X chromosome and autosomes at a significance level p < 0.05; half of the traits were affected by the Y chromosome and the species identity of the Y chromosome (a male parent identity); and one-third of the traits, by the interaction of the Y chromosome and autosomes.

Criteria for the size of the effect show the greatest influence of the autosomes on the total variability of the traits, and the least influence of the Y-chromosomes (Table 5).

Note that a dependence on the hereditary factors was not confirmed for the apodeme width-related traits (F1). The apodeme declination and curvature (F2) depend on the identities of the Y chromosome and the species identity of the Y chromosome (a male parent). Distinct species-specific traits of the *aedeagus* and parameres (F3) depend on the X chromosome, autosomes, and male parent identity. The shape of the ventral bend in the proximal part of the *aedeagus* outline (F4) is determined by the interaction between the Y chromosome and autosomes. Traits related to the height of the dorsal bend in the *aedeagus* outline (F5) and the shape of the hook and the outline bend over the hook (F6) showed a similar dependence on the Y chromosome, autosomes, and a male parent identity; an additional dependence on the X chromosome is specific to F5-related traits. Traits describing the shape of the distal part of the outline (F7) depend on the autosomes, male parent identity, and their interaction.

An independent effect of the Y chromosome is expected to be insignificant based on the composition and functional activity of its coding sequences. It is possible to assume that interactions of the two factors play a crucial role in the phenotype.

## The role of the components of variability and their combinations in the inheritance of traits of the shape of the male copulative organ

*Post-hoc* tests were used to evaluate the probability of the formation of homogeneous groups in trait variance according to a published model [59] (S5 Table) to detail the strength and direction of the effects exerted by the hereditary factors and their combinations on the dominance of the *D. virilis* phenotype. Criteria for the size of the effect of these factors show an approximately equal contribution of each of them to the total variability of the traits (S6 Table). The equal contribution indicates similar participation of the sex chromosomes,

**Table 6. Paired permutation test with the Bonferroni correction to evaluate the optimal genotype partitioning into groups homogeneous in latent variables and highly specific variables (up to 100 000 iterations in each case).**

| F# | | X→Y (epist) [A₁] | AUT (add) [A₂] | P→AUT(add) +P→Y [A₃] | Y→AUT (rec.epist) + X→AUT (rec.epist) [A₄] | Y+Y→AUT(dom. epist) [A₅] | P→AUT(dom) +P→X[A₆] | X→AUT (dom.epist) +X +AUT (dom) [A₇] |
|---|---|---|---|---|---|---|---|---|
| F1 | Prob. | 0.493 | 0.476 | 0.302 | 0.638 | 0.136 | 0.473 | 0.304 |
| | Dom. | n.s. | n.s. | n.s. | n.s. | n.s. | n.s. | n.s. |
| F2 | Prob. | 0.113 | 0.207 | **0.000** | 0.057 | **0.038** | **0.000** | **0.000** |
| | Dom. | n.s. | n.s. | $^*$(X-chr) | n.s. | + | +sD$^{Lu}$X-chr | +$^*$(sD$^{Vi}$X-chr) |
| F3+ | Prob. | **0.001** | **0.000** | **0.000** | **0.000** | 0.285 | **0.000** | **0.000** |
| | Dom. | + | +D$^{Lu}$ | + P_Y, D$^{Lu}$Aut | $^*$(sD$^{Vi}$Aut) | n.s. | +$^*$ | +$^*$sD$^{Lu}$X-chr |
| F4 | Prob. | 0.697 | 0.489 | **0.000** | 0.738 | **0.000** | **0.000** | **0.000** |
| | Dom. | n.s. | n.s. | +P_Y | + n.s. | + | +$^*$ | +$^*$sD$^{Lu}$X-chr |
| F5 | Prob. | **0.000** | **0.000** | **0.001** | **0.000** | **0.025** | 0.043 | 0.051 |
| | Dom. | + | +$^*$ | +$^*$ D$^{Lu}$Aut | + ID | + | n.s. | n.s. |
| F6 | Prob. | 0.478 | **0.000** | **0.000** | 0.857 | **0.000** | **0.000** | **0.000** |
| | Dom. | n.s. | + D$^{Vi}$ | + D$^{Vi}$Aut (X-chr) | n.s. | + | + D$^{Vi}$Aut | + ID |
| F7 | Prob. | 0.542 | **0.000** | **0.002** | 0.028 | **0.031** | 0.045 | **0.001** |
| | Dom. | n.s. | + sD$^{Vi}$ | +$^*$ D$^{Vi}$Aut | +$^*$ | + | n.s.(sD$^{Lu}$X-chr) | + D$^{Vi}$Aut |
| FDR | 0.05 | 0.021 | 0.043 | 0.036 | 0.036 | 0.043 | 0.029 | 0.021 |
| | 0.01 | 0.004 | 0.009 | 0.007 | 0.004 | 0.006 | 0.006 | 0.004 |
| Imp 3 | Prob. | **0.000** | **0.000** | **0.000** | **0.000** | 0.009 | 0.811 | 0.123 |
| | Dom. | + | + D$^{Lu}$ | + D$^{Lu}$Aut | + D$^{Vi}$X-chr | + | n.s. | n.s. |
| Imp 18 | Prob. | **0.000** | **0.000** | **0.009** | **0.000** | 0.008 | 0.222 | 0.085 |
| | Dom. | + | + D$^{Lu}$ | + P_Aut | + D$^{Vi}$X-chr | + | n.s. | n.s. |
| Imp 19 | Prob. | **0.000** | **0.000** | **0.002** | **0.000** | **0.000** | 0.302 | **0.009** |
| | Dom. | + | + ID | + D$^{Vi}$Aut | + ID | + | n.s. | + D$^{Vi}$Aut |
| Imp 23 | Prob. | 0.385 | **0.004** | **0.002** | **0.000** | 0.412 | 0.033 | **0.002** |
| | Dom. | n.s. | + D$^{Vi}$ | +$^*$ D$^{Vi}$Aut | +$^*$ | n.s. | n.s. | + ID |
| Imp 24 | Prob. | **0.010** | **0.004** | **0.004** | **0.003** | **0.000** | **0.000** | **0.016** |
| | Dom. | + | + D$^{Vi}$ | + D$^{Vi}$Aut | + ID | + | + D$^{Vi}$Aut | +$^*$ sD$^{Vi}$Aut |
| Imp 26 | Prob. | 0.055 | **0.000** | 0.145 | **0.000** | 1.000 | 0.239 | 0.120 |
| | Dom. | n.s. | + D$^{Lu}$ | n.s. | + D$^{Vi}$X-chr | n.s. | n.s. | n.s. |
| Imp 27 | Prob. | 0.306 | **0.000** | **0.004** | **0.000** | 0.253 | 0.054 | **0.002** |
| | Dom. | n.s. | + D$^{Lu}$ | +$^*$X-chr | +$^*$ sD$^{Vi}$X-chr | n.s. | n.s. | + D$^{Lu}$X-chr |
| Imp 28 | Prob. | **0.020** | 0.538 | **0.002** | 0.042 | 1.000 | **0.000** | 0.067 |
| | Dom. | + | n.s. | + D$^{Lu}$Aut | n.s. | n.s. | + D$^{Lu}$X-chr | n.s. |

Designations of the factors—as in Table 3. +—a significant effect of the hereditary factor associated with a linear relationship with the given classes of genotypes; +$^*$—a significant effect of the hereditary factor associated with non-linear dependence with the given classes of genotypes; (s) D$^{Vi (Lu)}$–dominance (overdominance) of the *D. virilis* (*D. lummei*) phenotype in genotype classes with an intermediate value of the indicator variable; PY (Aut)—nongenetic effect of the father origin factor on the Y chromosome (autosomes); Aut, X-chr, Y-chr—autosomes, X- or Y-chromosome, which determine the dominance of the phenotype. F#—the number of the latent trait, according to Table 5; F#$^*$—permutation test values according to the values of the factors (latent traits). CH—signs with a high characteristic. FDR—false discovery rate for the latent traits. (X-chr)—unaccounted influence of the X chromosome on the genotype of males $F_1$ X$_{Lu}$Y$_{Vi}$ under the influence of the hereditary factor P→Y + P→AUT(add) (explanation in the text); (sD$^{Vi}$X-chr (Aut))—changes in the phenotype of a latent trait in classes with intermediate and maximum values of indicator variables coincide but are statistically significant only for the former. Prob.–probability, Dom.–domination.

autosomes, and the species identity of the Y chromosome (a male origin factor) in the formation of the analyzed interactions.

The results obtained for each particular trait are provided in Supplementary; data on latent and highly specific traits are summarized in Table 6.

The model implies that each hereditary factor exerts a discrete effect on the traits in question, depending on the genotype. The expected effect of each factor was analyzed using the indicator variables listed in Table 6.

The results of analyzing how the primary and latent traits varied under the influence of the hereditary factors show a dependence on the hereditary factors that was observed for the majority of the latent traits and the corresponding primary traits with a high commonality. Trait F1, which includes traits related to apodeme width, was an exception; i.e., its dependence on any of the hereditary factors was not confirmed.

The epistatic effects exerted by the conspecific sex chromosomes and by the interaction between the Y chromosome and dominant alleles of the *D. virilis* autosomes positively affect the expression of the *D. virilis* phenotype in the majority of the morphological structures (F2 and F4-F7, IMP 3, 18, 19, 24 with high specificity).

A significant influence of the additive effect of recessive autosomal alleles (AUT(add)) was demonstrated for latent traits F3 and F5–F7 and confirmed for the majority of the corresponding primary traits and highly specific IMPs 3, 18, 19, 24, 26, and 27. Traits of the genotypes that were heterozygous for the autosomes and had an intermediate value of the indicator variable displayed differentiation by the dominance of one of the parental phenotypes. Wherein, latent trait F3, which characterizes the most distinct species-specific traits, and highly specific IMPs 3, 18, 26, and 27 showed the dominance of the *D. lummei* phenotype in heterozygotes for the autosomes. The additive effect of the autosomes significantly influences the dominance of the *D. virilis* phenotype and incorporates a dominant component that determines the expression of one of the parental phenotypes, depending on the trait in question. The dominant effect of the *D. lummei* autosomes is manifested for pronounced species-specific traits.

The epistatic effects of the sex chromosomes on recessive autosomal alleles (factor Y→AUT(rec.epist)+X→AUT(rec.epist)) significantly influence latent traits F3, F5, and F7 and highly specific primary IMPs 3, 18, 19, 23, 24, 26, and 27. Wherein, the epistatic effects that the *D. virilis* sex chromosomes exert on recessive alleles of the conspecific autosomes lead to the dominance of the *D. virilis* phenotype.

A combined effect of autosomal dominant alleles, the X chromosome, and their epistatic interactions (factor X→AUT(dom.epist)+X+AUT(dom)) was confirmed for the majority of the primary traits having a high commonality; latent traits F2–F4, F6, and F7; and highly specific IMPs 19, 23, 24, and 27. However, the direction of dominance differed among the traits. Latent traits F3 and F4 showed a dominance of the *D. lummei* phenotype in males (genotype F1 $X_{Lu}Y_{Vi}$), suggesting a predominant effect for the *D. lummei* X chromosome and autosomes. In contrast, traits related to the apodeme bend and declination (F2) showed the possibility of superdominance of the *D. virilis* phenotype in the intermediate genotype and a leading role of dominant alleles of the *D. virilis* autosomes. A similar effect of *D. virilis* autosomal dominant alleles was confirmed for latent trait F7 and highly specific IMPs 19 and 24 at high significance. Latent trait F6 and highly specific IMP 23 displayed incomplete *D. virilis* phenotype dominance of genotype $X_{Lu}Y_{Vi}Aut_{Lu/Vi}$.

The influence of the male parent identity on the effect of the X chromosome and autosomal dominant alleles (factor P→X+P→AUT(dom)) was demonstrated for latent traits F2, F3, F4, and F6 and highly specific primary IMPs 24 and 28. Latent traits F3 and F6 nonlinearly depended on the interaction of the male parent identity and the X chromosome and autosomes. The dominance of the *D. lummei* phenotype was observed in the genotype having the intermediate value of the indicator variable for latent trait F2; and highly specific primary IMP 28. A significant effect of this factor was not demonstrated for latent traits F1, F5, and F7. In contrast, individual primary traits incorporated in the latent factor displayed, in the intermediate genotype, the dominance of the *D. virilis* phenotype in the case of F1 and F5 and the

*D. lummei* phenotype in the case of F7. With F7, the mean values gradually increased in the order of genotypes with indicator variable values 1–0–2, suggesting superdominance of the *D. lummei* phenotype.

The effect of the father's origin on the action of the Y chromosome and recessive alleles of autosomes (factor AUT (add) _P + Y_P) is shown for all latent variables, except the first, and all primary characters with a high characteristic, except IMP 26. The Y chromosome, under the influence of the father's origin, makes a maximum contribution to the phenotype manifestation for the latent traits F3 and F4. Wherein, for the F3 latent trait, a mixed effect of the Y chromosome and autosomes is observed, leading to the dominance of the *D. lummei* phenotype in genotypes with intermediate values of indicator variables. The primary traits defining the latent trait F4 confirm the noted dependence. The result of the non-random distribution of genotypes into heterogeneous groups for the latent trait F2 and the primary trait with high characteristic IMP 27 does not allow us to unambiguously relate the distribution data to the influence of the estimated factor. In both cases, a key role is played by the independently clustered male genotype from F1 $X_{Lu}Y_{Vi}Aut_{Lu/Vi}$, which has the *D. lummei* X chromosome and, accordingly, all its effects, which suggests a possible contribution of this chromosome. The primary characters that make up F2 also show different variants of dependence on autosomes and Y- and X-chromosomes. Latent traits F5, F6, and F7, as well as primary traits with a high characteristic IMP 3, 18, 19, 23, 24, and 28, show a significant contribution of autosomes modified by the effect of the father's origin.

Thus, the estimates confirm that epistatic interactions of the sex chromosomes and autosomes and effects of the male parent origin from interspecific crosses influence the expression of *D. virilis* species-specific traits in the shape of the male copulatory system.

## Discussion

How do traits determining the shape of the male copulatory system become a target of selection? As mentioned in the Background section, the efficiency of female insemination in insects depends on how well the female and male genitalia match each other [52]. Incomplete insemination must make repeated mating more likely, lead to displacement of sperm from the previous mating, and facilitate efficient selection against the incomplete insemination-associated genotype in *Drosophila*. Hoikkala and colleagues [60] have analyzed the within-population variation of mating duration in *D. montana* and associated the duration of the first mating with female resistance to repeated mating. We have shown that sensory microchaetae on the ventral surface of the *aedeagus* mediate the association between evolutionarily significant parts of the *Drosophila* copulatory system with the duration of mating [61].

Both the copulation duration and the shape of the copulatory system are not directly involved in adaptation but are maintained close to the adaptive optimum of a population as a result of apparent stabilizing selection [62,63]. In other words, the selection at these characters acts through adaptively valuable traits characteristic of a particular group of individuals. This selection takes the form of sexual selection because the characters are specifically expressed as predictors of sexual reproduction efficiency. When a new adaptation forms and the adaptive norm changes rapidly, selection changes to directional or disruptive, but remains indirect. It is, therefore, possible to expect that the variation observed experimentally will correspond to the variation in quantitative traits affected by one of the above selection types.

Two models describe the variation maintained by stabilizing selection equally well: a joint-effect model and an infinitesimal model. In the former, the variation is associated with the effects of moderate-frequency alleles, which are nearly neutral in terms of fitness and substantially influence the trait in question. In the latter, the variation involves many genes that each

exert a minor effect on fitness and act additively. Modeling of comprehensive experimental data has shown that well reproducible results are obtained with both of the models, neither of them is preferable to the other [63]. An analysis of QTLs for commercially valuable traits in farm animals supports well the conclusion that recessive variation maintained in a population plays an important role [64–66]. The majority of the traits are related to morphological and physiological characteristics affected by stabilizing selection. The facts that a variety of farm animal breeds formed rapidly on an evolutionary scale, that homologous haplotypes are responsible for similar traits in different breeds and are present in populations of founder species, and that epistatic interactions are observed between QTLs indicate that diversity at quantitative traits is maintained because neutral variation is preserved in populations.

Lower estimates could be obtained for genetic variation at traits under stabilizing selection when the effects of suppressing epistasis are underestimated, the fact being masked by overestimating the effect of stabilizing selection and/or underestimating the magnitude of variation due to mutation [67,68]. Moreover, opposite epistatic effects may occur between tightly linked genes in a QTL [69] to reduce their total effect observed, and opposite effects occurring between different QTLs substantially increase the total variance.

The formation of evolutionary new and usually adaptively significant traits has other genetic dynamics. Fisher [70] has noted that adaptation is not selection, meaning fitness-based selection that eliminates the least fit individuals. Adaptation is characterized by the effect of positive selection, which facilitates the progress of a population to an optimal phenotype. Based on Maynard Smith's model of adaptive walk in sequence space [71] and Gillespie's theory of fixation of rare beneficial alleles [72,73], it is thought now that adaptations arise in bursts alternating with periods of slow evolution [74]. Adaptation is associated with a few effects, and selection rapidly decreases in strength with each subsequent step; i.e., an exponential distribution is characteristic of the selection coefficients of consecutively fixed mutations [75]. An allele that expands the ecological niche for the species will be fixed rapidly in a new QTL. Experiments on QTL mapping support the conclusions based on model analyses [76–79]. It is safe to say that, along with additive effects, dominant interactions of alleles gain principal importance. A detailed analysis of the genetic architecture and ontogenetic mechanisms of species-specific traits in two *Labeotropheus* species from the Malawi Lake has made it possible to evaluate the role that regulatory, or epistatic, interactions of genes involved in the Tgfβ signaling pathway play in the formation of foraging adaptations [80].

Additive, dominant, epistatic, and nongenetic components of variation underlie the phenotypic differences in our experiment. The experiment was not designed to determine the variance components precisely. However, given that the homozygous or heterozygous autosome sets were identical in the males examined, it is clear that changes in the dominance of the parental phenotypes are associated with the nongenetic effects of the male parent identity and epistatic effects of the X and Y chromosomes. Neither primary nor latent traits remained phenotypically the same in genotypes with different combinations of the sex chromosomes and different male parent identities, while the autosome set was identical.

The effects of the sex chromosomes and autosomes were specific to different groups of traits. For example, the additive effect of recessive autosomal alleles determined the traits of the *aedeagus* (apart from those characterizing the ventral bend in the proximal part of its outline) and parameres. A combined effect of autosomal dominant alleles, the X chromosome, and its dominant epistatic interaction with the autosomes was observed for traits of the *aedeagus* and parameres and the bend and declination of the apodeme. It is of interest that the two factors exerted similar effects in genotypes with intermediate values of indicator variables. One of these was the dominance of the *D. lummei* phenotype in latent trait F3, which combined the most distinct species-specific traits; certain primary traits that were incorporated in latent

traits F4 and F5 and similarly showed species specificity [81]; and paramere width (IMP 27). The dominance of the *D. virilis* phenotype was characteristic of latent trait F7; several primary traits were incorporated in latent traits F5 and F6, and paramere width in the ventral bend region (IMP 24). The examples show that the dominant component of the autosomes contributed to the effect of additive interactions and that its contribution depended on the epistatic interaction with the X chromosome. Following Huang and Mackay [82], the components of variance are impossible to strictly extracted isolate in the majority of cases, especially when the experimental design is not an orthogonal one, which at least formally ensures uncorrelated variance components. Our estimates are, therefore, the total effects of several components of variance. Minimization of the autosomal effect in the case of dominance of the *D. lummei* phenotype and, oppositely, minimization of the effect of the X chromosome and its nongenetic influence on autosomal dominant alleles show that the X chromosome plays alternative roles in the variation of different traits. Similar dominance changes observed for genotypes heterozygous for the autosomes when estimating the role of the additive component of variance confirms that the dominant component of the autosomes contributes to the species-specific variation.

The effect that epistatic interactions between the X and Y chromosomes exert on species-specific traits was confirmed for half of the primary traits and latent traits F3 and F5, which incorporate taxonomically significant traits. It should be noted here that unequivocal interpretation is impossible for the results indicating that the *D. virilis* phenotype is enhanced in males with conspecific sex chromosomes. Given that different variance components may contribute to this phenomenon, an independent effect can be expected for genes of the *D. virilis* X chromosome. For example, Carson and Lande [83] have shown that the formation of an evolutionarily new secondary sex characteristic (an additional row of bristles on the male tibia) is determined to the extent of 30% by sex-linked genes in a natural *Drosophila silvestris* race. As for the traits that contribute to isolating barriers, the most detailed data are available for sterility-related traits. The examples below illustrate interspecific hybrid male sterility as a model of interactions between autosomes and the sex chromosomes. A cluster of Stellate sequences on the *D. melanogaster* X chromosome codes for a homolog of the β subunit of protein kinase CK2 and its overexpression causes male sterility. RNA interference mediated by Y-linked Su (Ste) repeats prevents Stellate overexpression [84,85], demonstrating that male fertility strongly depends on the interaction between the X and Y chromosomes. A similar model of fertility regulation utilizes the Odysseus-site homeobox protein (OdsH), which is encoded by an X-chromosomal locus in species of the *melanogaster* group. OdsH binds to heterochromatin sequences of the Y chromosome. In many cases, the presence of heterospecific X and Y chromosomes leads to decondensation of Y-chromosomal heterochromatin, dramatically distorts the expression of autosomal genes, and causes sterility [24,86]. The genetic system also illustrates the interaction between the X and Y chromosomes.

The effect of epistatic interactions between the sex chromosomes and recessive alleles of the autosomes was confirmed for the majority of latent and primary traits. A leading role of epistatic interactions of the X chromosome is evident from the observation that the *D. virilis* phenotype dominated at the majority of traits in males with genotype $X_{Vi}Y_{Lu}Aut_{Vi}$. A substantial role of epistatic interactions between the X chromosome and autosomes has similarly been demonstrated for another trait involved in prezygotic barriers, namely, a species-specific male courtship song pattern in *Drosophila* species of the *montana* phylad [38]. There is evidence that the sex chromosomes exert a significant regulatory effect on the expression activity of autosomal genes. Expression of X-chromosomal genes in hemizygous *D. simulans* males depends on trans regulations to a substantial extent [87], and trans effects and joint action of

cis and trans effects of the X chromosome on genome-wide expression activity are the second most important to trans-regulatory effects of chromosome III in *D. melanogaster* males [88].

The important role that the Y chromosome plays in regulating the phallus shape-related traits in interspecific hybrids is possible to associate with trans-regulatory activity directly. The Y chromosome harbors only 23 single-copy protein-coding genes, and 13 of them are strongly restricted to the testis in expression and are mostly associated with hybrid male sterility [89–92]. Additive variation of the ten other genes should make a vanishingly small contribution to the phenotype at the quantitative traits of morphological structures. However, epistatic interactions of Y-chromosomal sequences with the X chromosome and autosomes have received experimental support. For example, experiments were performed to evaluate the activity of the male-specific lethal proteins/roX1,2 RNA complex, which is responsible for the dosage compensation of X-linked genes in *Drosophila* males. The Y chromosome proved to affect the viability in roX1, roX2 mutant males, the effect depending on the Y-chromosome source [93]. The viability was low in males with the paternal Y chromosome and high in males with the maternal X chromosome. The result indicates that the Y chromosome modifies dosage compensation through roX1, roX2-mediated modification of heterochromatin [94] and/or recognition of the X chromosome by the entire male-specific lethal proteins/roX1,2 RNA complex. The results support our finding of a substantial paternal effect, which acts independently in the majority of traits or combination with the X-chromosome effect in some other cases. Finally, Lemos and colleagues [92,95] have directly demonstrated the regulatory activity of the Y chromosome in a study of differential genome expression activity under the Y-chromosome influence.

Traits of the apodeme, a muscle attachment site, and an internal part of the copulatory system, are the least associated with evolutionary variation and chromosome effects. It is noteworthy that chromosome effects are mediated by the nongenetic effect of male parent identity in the apodeme traits. Note that significant nongenetic effects were similarly observed for all other groups of correlated traits. A change in the type of dominance of half of the analyzed characters in males with an identical genotype, differing only in the status of the father's origin, is the basis for identifying these effects as an independent factor. Neither the mitochondrial genome nor the genotype for the nuclear genome *per se* is related to this effect. It can be assumed that it may be associated with the Meiotic Sex Chromosome Inactivation mechanism (MSCI) or, in a broader sense, meiotic silencing of unpaired chromatin/DNA (MSUC/MSUD) [21,96], which leads to heterochromatinization and inactivates the chromosome regions with altered meiotic synapsis in leptotene. The MSCI mechanism and its substantial evolutionary role in organizing the chromosomes and facilitating postzygotic isolation have been the matter of extensive discussion [21,23,96,97]. MSCI is typically compensated for by the higher transcriptional activity of male-specific genes in *Drosophila* because their strong promoters are overrepresented in the X chromosome [98,99].

On the contrary, the delay in forming of heterochromatin blocks is often associated with sterility in hybrids from interspecific crosses and the death of hybrid progenies; defects in mitotic chromosome segregation are observed hybrids during their embryo development [24,25,96]. Meiotic silence of unpaired heterochromatin and delayed condensation of heterochromatin blocks, leading to lethality during mitosis, are opposite processes, but the marking of such blocks may be of a general nature. Epigenetic signatures are preserved in chromosomes at post meiotic stages and may be inherited through generations [92,100]. Although the regulation of genome-wide epigenetic states is often associated with the Y chromosome [92,101,102], a formal role of the interspecific hybrid status is noteworthy in our case, the manifestation of which may be associated with a violation of the diverging chromosomes leading to a violation of the synapse of divergent chromosomes and causing the formation and further preservation

of noncanonical heterochromatin regions with altered expression activity. Thus, Argyridou and Parsch note a decrease in the average expression activity of genes of the X chromosome in somatic tissues compared with autosomal genes [103]. Although the authors conclude that this decrease is associated with the X chromosome being depauperate in genes with very high expression, even after removing overexpressed genes from the pool of analyzed genes, significant differences in the expression activity persist for intestinal tissues in males and females. The X chromosome enrichment with male-biased genes has also been shown for gene expression in the *Drosophila* brain [104], confirming the noted effect of selection as a factor in the evolution of sexual dimorphism.

Another potential mechanism should be noted, the action of which determines the effect of the father associated with the action of micro-RNA. Although the work was performed on a transcriptome from human tissue, this is the first study to show the accumulation of specific intergenic repeat-derived RNAs in spermatozoa [105], which can influence embryonic genome activation and determine differentiation processes at subsequent stages of development.

## Conclusions

All types of genetic interactions between the chromosomes were revealed in our study, and most of the shape traits of the male copulatory organ demonstrated the dominance of the *D. virilis* type. Epistatic interactions between the X and Y-chromosomes, as well as between the sex chromosomes and autosomes, affect species-specific traits. The two sex chromosomes shifted phenotype dominance in the conspecific direction.

The effects of sex chromosomes on the shape of the male mating organ is comparable with the effect of autosomes, even though the effect sizes of the former were inferior to the latter. The X-chromosome contains several times fewer genes compared with autosomes, whereas the Y-chromosome contains no more than 20 genes. This insignificant number of genes is incommensurable with the effect sizes of the sex chromosomes, which leads to the assumption that natural selection supported the accumulation of structural and regulatory elements on the sex chromosomes. Also, it can be assumed that sexual selection for specific male-biased genes associated with the shape of the male mating organ might prevent the loss of these genes on the X chromosome.

## Supporting information

**S1 Fig. Genotype distribution in the space of the first two principal components.** The genotypes are abbreviated as in Table 2. The chromosomes and paternal genotype are indicated in the following order: X chromosome, Y chromosome, autosomes, male parent identity.
(DOCX)

**S1 Table. Variation of the morphometric traits in the shape of the male copulatory system.** The sample size is indicated in parentheses. M, mean; σ, standard deviation. Superscripts in the first column indicate that the trait is incorporated with a high weight in the respective factor structure.
(DOCX)

**S2 Table. Dominance at traits of the system of copulatory shape as dependent on the sex chromosome composition in *D. virilis/D. lummei* hybrid males homozygous for the *D. virilis* autosomes.**
(DOCX)

**S3 Table. Significant effects and interactions as revealed by factorial MANOVA.** All four factors and their interactions were used as predictors; the 35 phenotypic traits, as independent variables.
(DOCX)

**S4 Table. Effects of the sex chromosomes, autosomes, and parental genotypes on trait expression.** Traits are grouped according to their maximal weights in the respective factors (Table 3). F, Fisher's test; p, the significance of effects of independent variables, including X, the X chromosome; Aut, the autosomes; ♂P, the paternal genotype; and ChrY*Aut, a combined effect of the Y chromosome and autosomes. HC, a group of the traits that were not incorporated in the factor structures with weights higher than |0.5|. Significance values $p < 0.05$ are in bold. In each group of traits determining the respective factor structure, the lowermost row shows the estimated effects of the independent variables on the given factor.
(DOCX)

**S5 Table. Permutational ANOVA to separate the genotypes into groups homogeneous in trait values.** (+), a significant effect of a hereditary factor that is in linear relationship with the given genotype classes; (+*), a significant effect of a hereditary factor that is in nonlinear relationship with the given genotype classes; $D^{Vi (Lu)}$, dominance of the *D. virilis* (*D. lummei*) phenotype in genotypes with the intermediate value of the indicator variable; $P_{Y(Aut)}$, an epigenetic effect of the male parent identity on the Y chromosome (autosomes); the subscripts Aut, X-chr, and Y-chr indicate that the autosome set, X chromosome, or Y chromosome determines phenotype dominance; F#, latent trait number as in Table 3; F#*, permutation test values for the factors (latent traits) used as variables; CH, highly characteristic trait; FDR, false discovery rate for the primary traits.
(DOCX)

**S6 Table. Homogeneous groups in latent trait ($F_n$) variance identified by the *post hoc* tests.**
(DOCX)

**S1 Text. Additional material for the last subsection "The role of the components of variability and their combinations in the inheritance of traits of the shape of the male copulative organ" of the Result section.**
(DOCX)

**S1 Dataset. Raw data.**
(XLSX)

## Acknowledgments

We mostly appreciate the comments and suggestions of Dr. Venera Tyukmaeva from CEFE—CNRS, Montpellier, France, who helped us significantly to improve the manuscript and with English edition, and we are grateful to Natasha Grigorian, who significantly improved our English.

The research was done using equipment of the Core Centrum and Drosophila stock collection (Registration Number #6868145) of the Institute of Developmental Biology RAS.

## Author Contributions

**Conceptualization:** Alex M. Kulikov.

**Data curation:** Alex M. Kulikov.

**Formal analysis:** Alex M. Kulikov, Dmitriy G. Seleznev, Oleg E. Lazebny.

**Investigation:** Alex M. Kulikov, Svetlana Yu. Sorokina, Anton I. Melnikov, Nick G. Gornostaev.

**Project administration:** Alex M. Kulikov.

**Software:** Dmitriy G. Seleznev.

**Supervision:** Alex M. Kulikov.

**Validation:** Dmitriy G. Seleznev.

**Visualization:** Anton I. Melnikov.

**Writing – original draft:** Alex M. Kulikov, Oleg E. Lazebny.

**Writing – review & editing:** Alex M. Kulikov, Svetlana Yu. Sorokina, Dmitriy G. Seleznev, Oleg E. Lazebny.

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
