## [Decision Letter · Decision Letter 0]

23 Oct 2020

PONE-D-20-28492

The effects of the sex chromosomes on the inheritance of species-specific traits of the copulatory organ shape in Drosophila virilis and Drosophila lummei

PLOS ONE

Dear Dr. Lazebny,

Thank you for submitting your manuscript to PLOS ONE. After careful consideration, we feel that it has merit but does not fully meet PLOS ONE’s publication criteria as it currently stands. Therefore, we invite you to submit a revised version of the manuscript that addresses the points raised during the review process.

Unfortunately, it took longer time than usual to obtain reviews. Please address all the comments of both reviewers. Especially pay attention to clarifying your approach to map the characters to chromosomes. A pipeline proposed by reviewer 2 will definitely help. 

We look forward to receiving your revised manuscript.

Kind regards,

Igor V. Sharakhov

Academic Editor

PLOS ONE

Journal Requirements:

Reviewers' comments:

Reviewer's Responses to Questions

**Comments to the Author**

1. Is the manuscript technically sound, and do the data support the conclusions?

Reviewer #1: Yes

Reviewer #2: Yes

2. Has the statistical analysis been performed appropriately and rigorously? 

Reviewer #1: Yes

Reviewer #2: Yes

3. Have the authors made all data underlying the findings in their manuscript fully available?

Reviewer #1: Yes

Reviewer #2: Yes

4. Is the manuscript presented in an intelligible fashion and written in standard English?

Reviewer #1: No

Reviewer #2: Yes

5. Review Comments to the Author

Reviewer #1: In this manuscript, Kulikov et al. investigate the effects of sex chromosomes and autosomes on the male copulatory organ shape in Drosophila virilis and D. lummei using quantitative genetics. With the help of interspecific crosses and backcrosses, they found that the species composition of sex chromosomes and autosomes contributes to male genital shape. The overall experimental design, analysis and conclusion are mostly sound. However, there are still many things need to be revised before resubmission.

1. For reviewing purpose, the manuscript should be well organized. I noticed that there are no numbered lines in this manuscript. In addition, figure legends are hard to follow since they located at different positions and I believe figure 1 legend is messed up with the main text. Please show your respects for the labor of reviewers starting from a good format of the manuscript.

2. The main concern for me is figure 2 and table 1, it is hard to link the factors and scheme of morphometric parameters. I suggested use different colored numbers in figure 2 to indicate corresponding factors. In addition, since you have organ images taken by electron microscope, you should provide the organ images with numbers from the scheme in it to clarify the real structure of male genital shape.

3. The quality of figure 1 is low. In addition, it is very confusing when you use one character to indicate two genotypes at the same time. Since you only have eight genotypes, I suggested you clarify their genotypes one by one without abbreviations.

4. In the main text, you have IMP35 and -2, which I did not find in Figure 2 and table 5.

5. In the discussion part, why did you mention MSCI? I doubted that MSCI plays roles in somatic tissues including male genital organ.

6. In the method part, you said “they carries the following recessive autosomal markers”, I think “they” only referred to individuals in D. virils. Please double check this. For the markers, since markers on chromosome 4,5,6 are all eye colors, how can you differentiate genotypes with different composition of eye color genes? It will be great if you use a scheme including chromosomes with markers on it to show you interspecies crosses and backcrosses.

7. Please clarify the age and the virginity of males in the method part.

8. There are many grammar mistakes, please check carefully.

Reviewer #2: In this study, Kulikov et al. used a set of crosses between D. virilis and D. lummei and applied the methods of quantitative genetics to assess the variability of the shape of the male copulatory organ and the effects of the sex chromosomes and autosomes on its variance. The study showed that the male genital shape depends on the species composition of the sex chromosomes and autosomes. Overall, the work is sound and the conclusions are solid. The manuscript is well-written especially I enjoyed reading introduction and discussion. However, I suggest rewriting the results to make them more understandable to readers who may not be familiar with the method. Please start with an introductory paragraph to the methods of quantitative genetics. Instead the authors, start with Table 1 that list Factors (without prior explanation of them) and numbers that don’t mean anything from the beginning. I suggest drawing a crossing scheme and a pipeline of all analysis, especially paying attention to how the obtained data led to the conclusions. The methods section can also be expanded and detailed.

6. PLOS authors have the option to publish the peer review history of their article (what does this mean?). If published, this will include your full peer review and any attached files.

Reviewer #1: No

Reviewer #2: No

---

## [Author Response · Author response to Decision Letter 0]

6 Dec 2020

Dear Dr Sharakhov,

On behalf of the all the co-authors of our manuscript I would like to thank you for the comments and the opportunity to resubmit the corrected version. We appreciate the reviewers’ comments and find them very helpful for improving our manuscript. Below we respond to them in detail and hope you find our corrections sufficient.

Reviewer 1

In this manuscript, Kulikov et al. investigate the effects of sex chromosomes and autosomes on the male copulatory organ shape in Drosophila virilis and D. lummei using quantitative genetics. With the help of interspecific crosses and backcrosses, they found that the species composition of sex chromosomes and autosomes contributes to male genital shape. The overall experimental design, analysis and conclusion are mostly sound. However, there are still many things need to be revised before resubmission.

We are happy that the reviewer finds our study sound and we address the comments below.

1. For reviewing purpose, the manuscript should be well organized. I noticed that there are no numbered lines in this manuscript. In addition, figure legends are hard to follow since they located at different positions and I believe figure 1 legend is messed up with the main text. Please show your respects for the labor of reviewers starting from a good format of the manuscript.

We deeply apologize for the inconvenience in reviewing our manuscript. We added the line and page numbers, and for further convenience, the Methods section was moved forward to the regular place between Introduction and Results. We also revised the Figure 1 legend (Lines 159-164).

2. The main concern for me is figure 2 and table 1, it is hard to link the factors and scheme of morphometric parameters. I suggested use different colored numbers in figure 2 to indicate corresponding factors. In addition, since you have organ images taken by electron microscope, you should provide the organ images with numbers from the scheme in it to clarify the real structure of male genital shape.

We thank the reviewer for a great piece of advice. We introduced colors into the scheme (Figure 2A) and added a general view of the male mating organ (Figure 2B) that links the scheme with the photos. The final part (Figure 2C) presents three photos of the male mating organ of D. virilis, D. lummei, and their hybrid from F1.

We moved the Methods section forward, but Figure 2 kept its number, because a new figure (Figure 1) with a scheme of crosses was added.

3. The quality of figure 1 is low. In addition, it is very confusing when you use one character to indicate two genotypes at the same time. Since you only have eight genotypes, I suggested you clarify their genotypes one by one without abbreviations.

Thank you, we find this advice very helpful. The quality of Figure 4 (formerly Figure 1) was improved and the genotype abbreviations were replaced with genotypes per se: they are listed in the caption below. We improved the explanation of genotype abbreviations in the text of the manuscript: the first letter stands for the X chromosome, the second letter stands for the Y chromosome, the third letter stands for autosomes, and the fourth letter stands for the parentage of the parent male: D. virilis, D. lummei, or an interspecies hybrid; A designates the genotype of D. virilis, B designates the genotype of D. lummei, H designates the D. virilis/D. lummei heterozygote for autosomes and the hybrid status of the parent male.

4. In the main text, you have IMP35 and -2, which I did not find in Figure 2 and table 5.

We have made the appropriate corrections for the reviewer’s comment. The hook angle (alpha) was mistakenly named IMP35, now it has been corrected.

IMP 2 is the only morphometric parameter included in the composition of two latent factors at once, and we used IMP 2 with the minus sign to designate this. We removed the minus sign from the figure and Table 3.

5. In the discussion part, why did you mention MSCI (male meiotic sex chromosome inactivation)? I doubted that MSCI plays roles in somatic tissues including male genital organ.

We have rewritten this paragraph (Lines 781-786, 794-797, 800, 803-815).

6. In the method part, you said “they carries the following recessive autosomal markers”, I think “they” only referred to individuals in D. virils. Please double check this. For the markers, since markers on chromosome 4,5,6 are all eye colors, how can you differentiate genotypes with different composition of eye color genes? It will be great if you use a scheme including chromosomes with markers on it to show you interspecies crosses and backcrosses.

Thank you for this useful comment, we believe that a scheme will make our design clearer for the readers. A scheme of crosses was added as Figure 1. It is possible to differentiate between genotypes using different compositions of eye color genes, and we provided a better description of the mutation phenotypes in the Methods section (Lines 145-153). We also checked the sentence pointed out by the reviewer, and can confirm that the markers refer to D. virilis individuals (Line 141).

7. Please clarify the age and the virginity of males in the method part.

All males used for their copulatory organ examination were virgin and seven days old. This clarification has been added to the text (Lines 172-173).

8. There are many grammar mistakes, please check carefully.

The text was double-checked for mistakes and typos.

Reviewer 2

In this study, Kulikov et al. used a set of crosses between D. virilis and D. lummei and applied the methods of quantitative genetics to assess the variability of the shape of the male copulatory organ and the effects of the sex chromosomes and autosomes on its variance. The study showed that the male genital shape depends on the species composition of the sex chromosomes and autosomes. Overall, the work is sound and the conclusions are solid. The manuscript is well-written especially I enjoyed reading introduction and discussion.

Again, we are pleased that the reviewer found our study solid and interesting. We address further comments below.

1. However, I suggest rewriting the results to make them more understandable to readers who may not be familiar with the method. Please start with an introductory paragraph to the methods of quantitative genetics. Instead the authors, start with Table 1 that list Factors (without prior explanation of them) and numbers that don’t mean anything from the beginning.

We reorganized the Methods and Results sections and added some text for clarification (Lines 145-153, 159-164, 172-173, 322-324, 327-330, 363-370, 397-403). We also added the introductory paragraph before we started to describe statistical methods in the subsection “Analysis of morphological structures” (Lines 183-202).

2. I suggest drawing a crossing scheme and a pipeline of all analysis, especially paying attention to how the obtained data led to the conclusions.

We thank reviewer for this comment. We added figures with a crossing scheme and pipeline of all analyses (Figure 1 – Lines 159-164 and Figure 3 – Lines 314-318).

3. The methods section can also be expanded and detailed.

We added more detail in the Methods section (Lines 145-153, 155-164, 172-173, 185-204, 222-230, 314-318).

---

## [Editor Report · Decision Letter 1]

8 Dec 2020

The effects of the sex chromosomes on the inheritance of species-specific traits of the copulatory organ shape in Drosophila virilis and Drosophila lummei

PONE-D-20-28492R1

Dear Dr. Lazebny,

We’re pleased to inform you that your manuscript has been judged scientifically suitable for publication and will be formally accepted for publication once it meets all outstanding technical requirements.

Kind regards,

Igor V. Sharakhov

Academic Editor

PLOS ONE
---

## [Editor Report · Acceptance letter]

14 Dec 2020

PONE-D-20-28492R1 

The effects of the sex chromosomes on the inheritance of species-specific traits of the copulatory organ shape in *Drosophila virilis* and *Drosophila lummei*

Dear Dr. Lazebny:

I'm pleased to inform you that your manuscript has been deemed suitable for publication in PLOS ONE. Congratulations! Your manuscript is now with our production department. 

Kind regards, 

on behalf of

Dr Igor V. Sharakhov 

Academic Editor

PLOS ONE